# Current Progress in Natural Degradation and Enhanced Removal Techniques of Antibiotics in the Environment: A Review

**DOI:** 10.3390/ijerph191710919

**Published:** 2022-09-01

**Authors:** Shimei Zheng, Yandong Wang, Cuihong Chen, Xiaojing Zhou, Ying Liu, Jinmei Yang, Qijin Geng, Gang Chen, Yongzhen Ding, Fengxia Yang

**Affiliations:** 1College of Chemistry and Chemical and Environmental Engineering, Weifang University, Weifang 261061, China; 2Department of Pediatrics, Weifang People’s Hospital, Weifang 261041, China; 3College of Environmental Science and Engineering, Nankai University, Tianjin 300071, China; 4Agro-Environmental Protection Institute, Ministry of Agriculture and Rural Affairs, Tianjin 300191, China

**Keywords:** antibiotics, biodegradation, enhanced removal, advanced oxidation processes, membrane filtration, constructed wetland, microbial electrochemical systems, hybrid technology

## Abstract

Antibiotics are used extensively throughout the world and their presence in the environment has caused serious pollution. This review summarizes natural methods and enhanced technologies that have been developed for antibiotic degradation. In the natural environment, antibiotics can be degraded by photolysis, hydrolysis, and biodegradation, but the rate and extent of degradation are limited. Recently, developed enhanced techniques utilize biological, chemical, or physicochemical principles for antibiotic removal. These techniques include traditional biological methods, adsorption methods, membrane treatment, advanced oxidation processes (AOPs), constructed wetlands (CWs), microalgae treatment, and microbial electrochemical systems (such as microbial fuel cells, MFCs). These techniques have both advantages and disadvantages and, to overcome disadvantages associated with individual techniques, hybrid techniques have been developed and have shown significant potential for antibiotic removal. Hybrids include combinations of the electrochemical method with AOPs, CWs with MFCs, microalgal treatment with activated sludge, and AOPs with MFCs. Considering the complexity of antibiotic pollution and the characteristics of currently used removal technologies, it is apparent that hybrid methods are better choices for dealing with antibiotic contaminants.

## 1. Introduction

Since the discovery of penicillin and its application on industrialization and commercialization, hundreds of antibiotics have been discovered and are utilized [1]. Currently used antibiotics fall into a number of different classes, including β-lactams, tetracyclines, macrolides, aminoglycosides, amphenicols, quinolones/fluoroquinolones, sulfonamides, lincosamides, glycopeptides, and polyether ionophores [2]. Antibiotics are widely used for the treatment of diseases, such as bacterial infections in humans and animals, greatly reducing the mortality and morbidity of infectious diseases caused by pathogenic bacteria [3]. In addition, antibiotics are also used as drugs and feed additives in the breeding industry where they not only prevent diseases but can also promote animal growth [4]. However, antibiotics are typically only partially metabolized in animals and humans, with the remainder being discharged into the environment either as whole compounds or metabolites through feces and urine [5]. Importantly, antibiotics in the environment may induce the generation and spread of antibiotic-resistant bacteria (ARB) and antibiotic resistance genes (ARGs), which can spread to organisms and humans through horizontal gene transfer either by direct contact or through the food chain, leading to widespread resistance in ecosystems [6]. Some ARBs and ARGs detected in the natural environment are listed in Table 1.

For the protection of public health, the European Commission has established maximum residue limits (MRLs) for antibiotics and other pharmacologically active substances in foods of animal origin [12,13]. In this regard, significant efforts have been made to develop robust analytical methods that combine techniques such as high-performance liquid chromatography (HPLC) coupled with mass spectrometry (MS) and liquid chromatography coupled with tandem mass spectrometry (LC–MS/MS) to monitor antibiotic residues in food and environmental material. The methods of antibiotic extraction from different substrates vary. For example, antibiotic residues in food are often difficult to detect as the residues are often matrix-associated, which can interfere with their analysis [14,15]. Current separation and detection methods are often based on chromatography or immune-affinity analysis. Gas chromatography (GC) and liquid chromatography (LC) are widely used for separating and detecting antibiotics. Conventional detectors such as ultraviolet (UV) or fluorescence detection (FD) often do not provide structural information for non-targeted compounds and present challenges for the simultaneous detection of multiple samples [16,17]. Although bioassays are often sufficiently sensitive and selective, they may be too specific for the quantification of multiple antibiotics [18,19]. Gas chromatography–mass spectrometry (GC/MS), gas chromatography–tandem mass spectrometry (GC/MS/MS), liquid chromatography–tandem mass spectrometry (LC/MS/MS), and high-performance liquid chromatography–MS/MS (HPLC/MS/MS) can make up for these limitations. HPLC/MS/MS currently dominates the detection methods for antibiotic residues in food and other environmental materials as most antibiotics are not volatile, therefore, not suitable for GC/MS and GC/MS/MS.

Antibiotic pollution is widespread in the environment and can be detected in surface water, groundwater, sediments, soil, and even drinking water [20,21]. The sources of antibiotic pollution are widespread and include pharmaceutical wastewater in which antibiotics are not completely degraded [22], human and veterinary antibiotics that have not been fully metabolized and large amounts of expired and unused antibiotics [23]. Sixty-eight different veterinary pharmaceutical residues were detected in natural water and tap water at concentrations ranging from ng/L to μg/L [24]. Even after a series of treatment processes in a drinking water plant, the effluent contained trace levels of antibiotics. A total of 58 antibiotics were found in the filtered tap water, with the mean and median concentrations of the detected antibiotics being 182 ng/L and 92 ng/L, respectively [25]. In addition, coastal waters in many parts of the world have also been contaminated with antibiotics, with more than 30 antibiotics detected in Chinese coastal waters at concentrations as high as μg/L [26]. For instance, the concentrations of 11 antibiotics in Bohai Bay ranged from 2.3 to 6800 ng/L [27], with concentrations of 12 antibiotics ranging between 0.1 and 90 ng/L in the Yangtze Estuary [28], indicating that large quantities of antibiotics have been released into coastal waters [29].

Antibiotics are even found in groundwater. The three main sources of antibiotic contamination of groundwater are drainage from the soil, infiltration from sediment, and introduction through the exchange between surface water and groundwater [30]. Antibiotics may enter groundwater through various agricultural activities such as aquaculture sewage irrigation and manure return, as well as runoff and rainwater leaching in aquaculture and agricultural areas [31]. There are several factors influencing antibiotic residuals entering to groundwater, including adsorptive, hydrogeological, physicochemical, and hydrochemical factors [31,32]. For example, because they are easily absorbed by sediments, tetracyclines are not common in aquatic environments, while sulfonamides have a high detection rate and are the most frequently reported antibiotic group in groundwater due to their weak adsorption capacity to soil and low biodegradability, which makes them easy to migrate into the water environment [33]. Aquifers’ geological features and hydrological sources contribute to its recharge and antibiotic content [31]. For example, antibiotics enter groundwater through lateral or longitudinal hydraulic exchange in stall belts [34]. In places where the surface water level is higher than the regional groundwater level, for example in arid areas, the unsaturated zone below the river channel is directly replenished by surface water. This mechanism is considered to be the source and path of groundwater pollution. Seasonal changes were observed in the type and concentrations of 19 antibiotics from 27 groundwater samples [35]. In spring, high concentrations of ofloxacin, norfloxacin, tetracycline, and erythromycin dehydrate were found in groundwater and surface, while in autumn, chlorotetracycline, doxycycline, and enrofloxacin comprised the greatest proportion with the highest concentrations in all the samples. Most of the sulfonamides were present at higher levels in spring than in autumn, which can be ascribed to surface runoff by rainwater during the wet season (spring). The average concentrations of compounds in the fluoroquinolone and tetracycline classes were far higher than those in the sulfonamide and macrolide classes. The main antibiotics present in groundwater were also the dominant compounds found in surface water, with correlation coefficients of 0.93 and 0.97 in autumn and spring, respectively, indicating the potential contamination of groundwater by the infiltration of contaminated surface water. In addition, the environment surrounding fertilized land, including surface water and groundwater, is affected at the same time due to the high concentrations of both antibiotics and ARGs in the fertilized soil which can be transmitted to the groundwater and surface water by runoff and irrigation [36].

Although antibiotics in the environment are usually detected only at the trace concentration level, “false persistence” may occur due to poor degradation of antibiotics and persistent input resulting from continuous discharge into the environment [37].

Residual antibiotics in the environment pose both toxicological effects and ecological risks [38]. For example, ciprofloxacin and sulfamethoxazole can inhibit the growth of algae in the ocean [39]. Trimethoprim reduces the abundance of hydra intestinal bacteria and also affects the digestion and absorption of algae by water fleas, ultimately affecting the growth of water fleas [40]. Low-mass concentrations of sulfadiazine leakage can cause increases in the autonomous movement and heart rate of the zebrafish, while exposure to drugs can lead to embryonic deformities [41]. Antibiotics also interact with microbes within the human body, resulting in imbalances in the gut flora, and causing various diseases [42,43]. Therefore, persistent antibiotic residue in the environment is a threat to both the structural stability of ecosystem and human health.

Accordingly, the environmental problems caused by antibiotics have become a significant challenge that requires urgent solutions and have attracted attention from environmental scientists around the world [1]. Although antibiotics can be degraded to a certain extent in the environment, this does not fully deal with the pollution problem of antibiotics. Consequently, the development of improved and efficient processes for antibiotic removal is an urgent problem for scientists. This review summarizes the mechanisms of antibiotic degradation in the natural environment as well as specific techniques for antibiotic removal, including hybrid techniques, to understand the progress of current technology and ultimately promote antibiotic removal.

## 2. Degradation of Antibiotics in the Natural Environment

Antibiotics can degrade to varying degrees in different environments by non-biological and biodegradation processes. Non-biological degradation includes photolysis, hydrolysis, oxidative degradation, and ionizing radiation degradation, while biodegradation in the natural environment includes degradation through the action of microbes, algae, and plants [44]. Table 2 lists the half-life times of antibiotics in different environment. The majority of half-life times are long, indicating that the degradation rates of antibiotics are generally low, and the antibiotics have characteristics of persistence in the natural environment.

### 2.1. Photolysis

Photolysis in the natural environment can be divided into three pathways, namely, direct, indirect, and self-sensitized photolysis [52].

Direct photolysis refers to the process by which the antibiotics absorb photons directly, causing the molecules to become excited through the transference of light energy and leading to the formation of products through bond breakage or structural rearrangement [53]. The process of direct photolysis is illustrated in Figure 1. This type of photolysis usually occurs in antibiotic molecules with groups capable of absorbing photons. It was found that the ecotoxicology of ciprofloxacin was reduced by direct photolysis at pH values of 5, 7, and 9 [54]. The degradation of nitrofuran by light was reported to occur in the same manner, with degradation occurring in two stages, namely, isomerization of the molecular structure after photon absorption and the subsequent fracturing of molecular bonds [55]. The first stage of the reaction was fast, while in the second stage, the breaking of the molecular bonds was rate limiting. Antibiotics such as quinolones and tetracyclines can undergo direct photolysis under ultraviolet irradiation at λ > 290 nm; the photolysis reaction follows pseudo-first-order reaction kinetics, and the formation of the equally potent of them leads to an increase in antimicrobial activity [56].

During indirect photolysis, light-absorbing materials existing in the environment (photosensitizer, such as oxygen, hydroxyl, and hydrogen peroxide) absorb light energy leading to excitation and the production of reactive oxygen species (ROSs) which then excite and degrade the antibiotics. The indirect photolysis process is also shown in Figure 1. The photosensitizer can alter the photostability of compounds to accelerate photolysis [57]. Strong photosensitizers can accelerate the photolysis of antibiotics by capturing free radicals (such as ·O_2_^−^ and ·OH) [58]. In natural water, this process is accompanied by the depletion of hydrogen ions, an increase in pH, an increase in the rate of deprotonation of radical cations, and a shift in the protonation equilibrium to deprotonated species [59]. Photosensitizers are widespread in nature, for example, dissolved organic matter ((DOM) such as humus, riboflavin), N_3_^−^, NO_3_^−^, NO_2_^−^, Fe^3+^, Fe^2+^, and NaCl. The O_2_ byproduct of Fe^3+^/Fe^2+^ interconversion can enhance the efficacy of ROSs (•OH and ^1^O_2_) induction by photosensitizers [60]. Cl^−^ competes with H_2_O_2_ for H_2_NO_2_^+^ to yield NOCl, which stimulates the production of ROSs (H_2_O_2_) by photosensitizers [61]. The production of various types of ROSs have been found in the surface water under the action of sunlight, where the concentrations of hydroxyl radical (•OH) and oxygen radical ^1^O_2_ were about 10^−17^~10^−15^ mol/L and 10^−15^~10^−12^ mol/L, respectively, and the antibiotic molecules can undergo rapid oxidative degradation in the presence of ROSs [62]. The study of Dai et al. showed that the direct photolysis efficiency of tetracycline was inefficient, and the indirect photolysis by photosensitizers was the principal means of degradation [58].

Self-sensitized photolysis occurs when antibiotics themselves absorb photons and become activated and subsequently transfer energy to other compounds while the antibiotics return to their original state. ROSs are generated during this process, and the antibiotics are degraded by reactions with ROSs [52,63]. To take ^1^O_2_ as an ROS example, as shown in Figure 2, the photosensitizer in ground state S_0_ absorbs light to produce a singlet excited state S_1_, which can decay non-radiatively by intersystem crossing (ISC) to a triplet state T_1_. Therefore, ^1^O_2_ would be produced after ^3^O_2_ obtains the energy of the triplet state, and the photosensitizer is returned to its original ground state S_0_. Studies have shown that antibiotics such as fluoroquinolones, tetracyclines, and chloramphenicol undergo not only direct photolysis, but also self-sensitized photolysis [64,65]. Natural organic matter (OM) and nitrates can photogenerate •OH. The contribution of •OH to self-sensitized photolysis was found to be positively correlated with the nitrate levels in natural water samples, demonstrating its important role in promoting self-sensitized photolysis of antibiotics. However, the contribution of OM to •OH generation in self-sensitized photolysis could not be clearly identified due to the relatively small differences in total organic carbon (TOC) levels in natural water [66]. 

Photolysis is an important contributor to antibiotic degradation in the environment [67]. Whether photolysis can occur depends largely on the chemical structure of the pollutant. Photolysis occurs mainly in the water and soil, and many environmental factors, including light intensity, environmental pH, catalysts, and metal ions, significantly impact antibiotic photolysis. Most current studies on photolysis focus on the reaction conditions, reaction rates, reaction substances, and their products involved in photolysis. For example, pH has a significant effect on the degradation of tetracycline since alkaline conditions are more favorable to photolysis and the rate of the reaction increases as the pH rises [65]. Different degradation pathways with different products are activated as the pH changes, illustrated by the photolytic degradation of enrofloxacin under different pH conditions [68] in Figure 3. Under ultraviolet irradiation at pH4, the F atoms of enrofloxacin are replaced by hydroxyl group (E1), and the piperazine side chain is oxidatively rearranged (E2 and E3). However, at pH8, the ethylpiperazine ring is oxidatively ruptured (E4), the F atoms are replaced, and the cyclopropane ring is oxidatively ruptured (E5 and E6). Batista et al. found that in the absence of Fe^3+^, the photolysis rates of sulfonazine and sulfonthiazole in the aquatic environment were only one-third of the rates in the presence of Fe^3+^ [69]. Iron can exist in the Fe(OH)^2+^ form in aqueous solution, and is converted to Fe^3+^ through a series of reactions under light with the production of ∙OH and HO_2_∙, thus promoting antibiotic photolysis as represented in the following reactions in Equations (1)–(5):Fe(III)-org complex + hν → [Fe(III)-org complex] * → Fe(II) + org radical(1)
org radical + O_2_→ O_2_^•−^ + oxidized org(2)
Fe(III) + 2H^+^ + 2O_2_^•−^→ Fe(II) + 2HO_2_•(3)
2HO_2_•↔ H_2_O_2_ + O_2_(4)
H_2_O_2_ + Fe(II) → Fe(III) + •OH + OH^−^(5)

### 2.2. Hydrolysis

Most antibiotics that can be hydrolyzed are water-soluble and the degree of hydrolysis is dependent on the specific characteristics of antibiotics [70]. Tetracycline, β-lactams, and macrolides are relatively easily hydrolyzed [71,72,73,74,75] while quinolones and sulfonamides are not [76,77]. In acidic conditions, the hydrolysis of tetracyclic antibiotics mainly proceeds though two different reactions [72,73]. One is the dehydration reaction between hydroxy group at C-6 and hydrogen on C-5 (Figure 4A(a)), and the other is the epimerization reaction of dimethylamine group at C-4 (Figure 4A(b)). While under alkaline conditions, the ring of tetracycline is opened to form lactone isomer owing to the nucleophilic attack of OH^–^ on C-11 (Figure 4A(c)) [72,73]. The hydrolysis process of β-lactam antibiotics is the break of amide bonds (ampicillin as an example, Figure 4B(c)) or the open of quaternary ring (Figure 4B(b)) [74]. For macrolide antibiotics, the glycosidic bonds among sugars break in acidic conditions, so do the glycosidic bonds between sugars and macrolide ring (spiramycin as an example, Figure 4C(b)), and in alkaline conditions the ester bonds in lactone ring hydrate, which can open the macrolide ring (Figure 4C(a)) [75].

The factors affecting antibiotic hydrolysis are mainly pH and temperature [71]. It was found that the degradation rates of chloramphenicol and penicillin in alkaline conditions were comparatively high, mainly due to their structures which can be attacked by electrophilic and nucleophilic substances [71]. It was reported that increasing the temperature accelerated the hydrolysis of oxytetracycline [78]. When hydrolysis temperature increased from 4 °C to 60 °C, the half-life of oxytetracycline was reduced from 120 d to 0.15 d, indicating that the hydrolysis rate increased by about 20% per 8 °C. The hydrolysis of oxytetracycline followed the first-order dynamics at different initial concentrations and was more likely to occur at neutral pH, followed by alkaline conditions. Hydrolysis was also promoted by increased temperature but was significantly reduced in the presence of Ca^2+^ [78]. A study by Loftin et al. demonstrated that the hydrolysis rate of tetracycline increased in relation to both pH and temperature, but was significantly reduced when the water conditions were similar to those of natural water [71].

### 2.3. Biodegradation

Biodegradation occurring under natural conditions contributes significantly to antibiotic degradation, with microbes, algae, and plants being largely responsible [79,80].

Microorganisms can change the structure and physicochemical properties of antibiotics and degrade antibiotic residues from macromolecular compounds to small molecule compounds and ultimately to H_2_O and CO_2_, thus rendering the antibiotic pollutants harmless. ARBs play important roles in antibiotic degradation. Both intracellular and extracellular enzymes produced by the ARBs can degrade and inactivate the antibiotic structure by mechanisms including hydrolysis, group transfer (such as acetyl transfer), and redox reactions [44]. Many microbial strains capable of degrading antibiotics have been identified in the environment; these include photosynthetic bacteria, lactic acid bacteria, actinomycetes, yeast, fermentation filamentous bacteria, *Bacillus subtilis*, and nitrated bacteria [81]. It was reported that a yeast strain isolated from the outlet sample of a pharmaceutical sewage plant degraded tetracycline to a rate of 78% [82]. Maki and colleagues isolated and obtained strains from sediment of cultured seawater fish that were capable of degrading both oxytetracycline and doxycycline efficiently [83]. The main environmental influences on microbial degradation of antibiotics include pH, temperature, oxygen content, and the environmental medium [84]. The use of microbial degradation to treat antibiotic pollution has the advantages of low cost and strong specificity.

Plant degradation refers to the degradation, transformation, absorption, metabolism, and detoxification of pollutants by plants to restore contaminated soil, water, and air [85]. As plants have large leaves and well-developed roots, they can carry out complex material and energy exchange, thus playing an important role in the ecological environment. Generally, there are three mechanisms involved in the plant degradation of antibiotics [86]. Firstly, the plants directly absorb organic pollutants and convert them into non-toxic metabolites that accumulate in plant tissues [87]. Kumara et al. found that corn (*Zea mays* L.), green onion (*Allium cepa* L.), and cabbage (*Brassica oleracea* L. Capitata group) can absorb chlortetracycline in the soil but had limited absorption and removal capability for tylosin [88]. Crop plants (*Cucumis sativus*, *Lactuca sativa*, *and Phaseolus vulgaris*) have been found to be able to metabolize enrofloxacin into ciprofloxacin and converted about one quarter of stored enrofloxacin [89]. Secondly, plants release secretions to degrade organic pollutants. Plants can degrade organic pollutants into less toxic or even harmless small molecules through a large number of organic acids and amino acids secreted by their roots; it can also provide a more suitable living environment for root microorganisms to promote their growth and reproduction through relevant biological processes such as root exudates and oxygen secretion, so as to improve the degradation ability of microorganisms [90]. Thirdly, plants promote the absorption, utilization, and transformation of organic pollutants by rhizosphere microorganisms. Hoang et al. used *Acrostichum aureum* L. and *Rhizophora apiculata* Blume Fl. Javae to repair the soil contaminated with quinolone antibiotics (ciprofloxacin and norfloxacin) and found that the process to degrade antibiotics was mainly through root microorganisms and the repair efficiency reached over 97% [91]. Chen et al. found that the rhizosphere biodegradation contributed to 90.2–92.2%, while hydrolysis (7.63–8.95%) and plant absorption (0.05–0.17%) were only auxiliary removal routes [92].

Currently, the studies of antibiotic degradation by algae are chiefly have focused on microalgae, and the mechanisms and studies on antibiotic removal by microalgae are in the Section 3.7.

## 3. Enhanced Removal Techniques

As current treatment cannot completely remove antibiotics, there are relatively high concentrations of antibiotic residues in the environment. At present, while urban wastewater treatment plants can effectively remove conventional organic pollutants and salts, antibiotics do not fall on the target list of pollutants to be removed, resulting in low removal efficiency [93]. Table 3 shows the antibiotic concentrations in the inflow and outflow of several wastewater treatment plants. It was found that the levels of ARBs and, especially, ARGs in urban sewage were significantly higher than those in natural or less affected water bodies [94].

As shown in Table 3, the concentrations of cephalexin, tetracycline, sulfamethoxazole, ciprofloxacin, trimethoprim, and sulfamazine were relatively high in the inflow of the wastewater treatment plant, with the highest levels reaching μg/L and the concentration of erythromycin-H_2_O even as high as mg/L. After treatment by the wastewater treatment plant, the highest concentration of several antibiotics in the effluent was less than 50% compared with the highest concentration in the inlet; these included cephalexin, sulfadiazine, norfloxacin, sulfamethoxazole, and lincolnensin, with the highest effluent concentration of cefotaxime being even higher than the highest concentration in the influent.

It can be seen that urban sewage treatment plants have become an important point source for the spread of antibiotic pollutants to the environment [100]. Therefore, there is an urgency to develop specific techniques for the treatment of these antibiotics. A number of techniques have been developed specifically for antibiotic treatment, including biological, chemical, and physicochemical methods.

### 3.1. Biotechnology

The application of biotechnology to the antibiotic pollution problem uses organisms (mainly bacteria and fungi) for degradation, and may also be accompanied by adsorption, hydrolysis, and photolysis [101]. The removal primarily depends on the degradation or transformation of antibiotics by anaerobic or aerobic microorganisms [102]. In general, the removal rates have been improved through domestication and optimization of microbial populations [103]. Liao and colleagues used mixed bacteria for the removal of sulfonamides and found that the principal bacteria in the sulfonamide antibiotic-degrading bacterial community were Firmicutes and Bacteroidetes (mainly the classes Bacilli and Flavobacteriia) [104]. Chen et al. found that under both aerobic and anaerobic conditions, the removal rates of nine antibiotics (sulfonamide monomexazine, sulfamidazine, sulfamadazine, trimethoprim, norfloxacin, ofloxacin, lincomycin, leucomycin, and oxytomycin) in piggery wastewater using an aeration biofiltration system were all greater than 82% [105]. Using a combined anaerobic–aerobic biological method for treating the piggery wastewater (mainly including sulfonamides and β-lactam), it was reported that anaerobic digestion mainly reduced chemical oxygen demand (COD) while the aerobic process contributed significantly to the antibiotic removal, leading to total COD and antibiotic removal rates of 95% (mainly attributed to anaerobic digestion) and 92% (mainly attributed to aerobic biodegradation), respectively [106]. These findings indicate that there are significant differences in the rates of antibiotic removal between anaerobic and aerobic microorganisms, and suggest that combining the two would be more effective.

Biological treatment is traditionally used for treating pollutants as it is cost-effective and conducive to large-scale operations. However, biological treatment does not completely remove antibiotics from the wastewater [107]. Due to the low concentrations of antibiotics in the environment, it is difficult to remove antibiotics by traditional treatment techniques because although traditional sewage treatment plants can remove large amounts of antibiotics in wastewater, they are unable to eliminate low concentrations of antibiotic residues. Moreover, hydrophobic antibiotic residues are prone to enrichment in the organic matter-rich sewage sludge [108]. Since the most commonly used removal mechanism for antibiotic residues and ARGs is adsorption, neither the antibiotic residues nor the ARGs are significantly damaged or degraded and continue to exist and accumulate in large quantities in the sludge. Therefore, it is necessary to explore more efficient wastewater treatment for antibiotic removal by combining biological treatment with other techniques.

### 3.2. Membrane Filtration

Membrane filtration can effectively remove macromolecular compounds including antibiotics [109]. The effectiveness of membrane filtration depends on the physicochemical properties of both the solution and compounds, as well as the material properties of the membrane [110]. At present, much research has focused on composite nanofiltration membranes, such as silver-mixed nanomaterials, multi-walled carbon nanotubes, and titanium dioxide [111]. Recent reports have shown that membrane filtration can also effectively remove ARGs in wastewater. Liang et al. applied a combination of ultrafiltration and two-stage reverse osmosis to treat piggery wastewater, which not only removed pollutants such as nitrogen and phosphorus but also trapped 72.64% of ARGs, effectively reducing the risk of ARGs to the natural water bodies [112].

Membrane filtration can encounter problems in practical applications from the presence of complex components in the wastewater, which can block filters and increase energy consumption. Furthermore, the concentrated effluent can easily contribute to secondary pollution if improperly treated. Thus, due to its expense and operating costs, membrane filtration is mostly used in drinking water treatment facilities and is rarely used for antibiotic wastewater treatment [113].

### 3.3. Adsorption

The adsorption method adsorbs the antibiotics in wastewater mainly by electrostatic interaction, hydrogen bonding, pore filling, and hydrophobic action [114]. Effective removal depends on the physical properties of the adsorbent material and the physicochemical properties of the solution and compounds. Some adsorbents require preactivation to increase their surface area and improve their adsorption rate. It was found that the removal rate of amoxicillin was higher than 80% using bentoxite modified by cetyltrimethyl ammonium [115]. Shao and colleagues synthesized MnFe_2_O_4_/activated carbon magnetic composite observing that tetracycline removal was strongly pH-dependent, decreasing from 95.2% at pH 3.0 to 64.6% at pH 11.0 [116].

Although adsorbents are effective for the removal of antibiotics under specific conditions, they have problems such as high cost, a need for regeneration when saturated, and are difficult to treat after use. To solve the cost problem, many researchers have focused on the preparation of low-cost modified adsorbents, such as biochar, corn bracts, and cellulose [117]. Zheng and colleagues studied the adsorption efficacy of biochar produced at 300–600 °C on the removal of sulfonamide antibiotics and found that adsorption depended on the pH, antibiotic concentration, and inorganic components of the biochar [118]. Rathod et al. synthesized crystalline nanocellulose using the green seaweed *Ulva lactuca* as raw material and demonstrated that the maximum adsorption amount of tetracycline hydrochloride was 6.48–7.73 mg/g [119]. A study by Yu et al. showed that corn bracts modified by zirconium ions can absorb 73 mg/g of levofloxacin from wastewater [120].

### 3.4. Advanced Oxidation Processes

Advanced oxidation processes (AOPs) are effective techniques used in wastewater treatment. The mechanism of removal is essentially based on the in situ generation of ∙OH, which reacts rapidly with most organic compounds (except chlorinated alkanes), although they lack selectivity of attack. ∙OH has a high redox potential and is capable of strong non-selective oxidation, and can thus oxidize various groups in the antibiotic molecular structure. The possible attack sites of ∙OH against several macrolides [121] are illustrated in Figure 5. Multiple factors can influence the treatment efficiency of AOPs, including the capacity of oxidant oxidation, the dose of oxidant, the action of the catalyst, the pH value, and the concentration of pollutants [122].

Many AOPs are based on a combination of strong oxidants (e.g., ozone, hydrogen peroxide, sulfate radicals (SO_4_^−^)), with catalysts (e.g., photocatalysts like perovskite), or radiation (e.g., ultraviolet or ultrasound). Suwannaruang et al. [123] used Ce_x_Sr_1−x_Fe_x_Ti_1−x_O_3_ perovskite catalysts for the enhancement of β-lactam antibiotics photodegradation under visible light irradiation. A diagram illustrating the degradation mechanism of the Ce_0.04_Sr_0.96_Fe_0.04_Ti_0.96_O_3_ photocatalyst as an example is provided in Figure 6. When the catalyst is irradiated with visible light, electron and hole pairs are formed in the materials. The holes in valence band of semiconductor can induce an oxidation reaction at +2.08 eV, while the electrons in conduction band of the materials are used for the reduction reaction at −0.42 eV. Superoxide anion radicals (O_2_•^–^) and H_2_O_2_ can be produced by the reaction of the adsorbed O_2_ with the photogenerated electrons on the conduction band. In addition, active OH• can be formed by subsequent reaction of the generated H_2_O_2_ with the photogenerated electrons, and OH• can be produced in the photogenerated holes on the valence band. Thus, the generated active OH• and O_2_^•–^ radicals can degrade the antibiotic amoxicillin into small structural intermediates or harmless products.

However, many other oxidants are selective and only react with some of groups in the antibiotic molecular. The findings of various studies [124,125,126,127,128,129,130,131,132,133,134] on the attack sites of O_3_, ∙OH, and ClO_2_ against sulfamethoxazole, tetracycline, ciprofloxacin, roxithromycin, and amoxicillin are shown in Figure 7.

Ostman et al. found removal rates of over 90% for ciprofloxacin, erythromycin, metronidazole, and trimethoprim using ozonation in a sewage treatment plant [135]. It was reported that the combination of ultraviolet (UV) and peroxydisulfate degraded sulfonamides more efficiently than the combination of UV and H_2_O_2_, with a removal rate of close to 99% [136]. Through studying UV/H_2_O_2_/O_3_, Lester et al. showed that the concentration of H_2_O_2_ was an important parameter in the two sub-processes of UV/H_2_O_2_ and H_2_O_2_/O_3_ and can fully mineralize ciprofloxacin and trimethoprim [137]. Two major reaction pathways for oxidative degradation of sulfasalazine by O_3_/H_2_O_2_ system [138] are displayed in Figure 8. OH first attacks the azo group (pathway 1) followed by the sulfonamide group (pathway 2), resulting in the rupture of N=N and N-S bonds, respectively (only some of the intermediates are shown in the figure). Lastly, the ROSs react further with these intermediates to produce end products, such as carbon dioxide, water, and inorganic ions [139]. The oxidative degradation of sulfasalazine clearly manifests that intermediates produced in the reaction are out of control and, as the toxic effects of these intermediates are unknown, this problem should be taken seriously. For example, while 73% of cefradine was degraded within 0.237 h by UV treatment, the toxicity of its byproducts was 1.04 times greater than that of the parent compound [140].

Despite their high removal efficiencies and rapid removal rates, AOPs cannot be applied to the large-scale removal of antibiotics from wastewater due to their high operational costs [141]. To address the feasibility of large-scale applications, improvements in processing costs, reductions in by-product toxicity, (optical) catalyst technology, and reactor design are needed.

### 3.5. Microbial Electrochemical Systems

Microbial electrochemical systems (MESs) are an emerging technology that uses microorganisms to generate electricity from chemical energy, and their use has received significant attention as an environment-friendly and cost-efficient method in recent years [142]. MESs exploit the catalytic activity of microorganisms to obtain electrons from available organic matter or inorganic substances. Depending on the partial potential in the MESs, the microorganisms can act as electron donors or electron acceptors. The microbial fuel cell (MFC) is the first and most studied MES prototype operated in the galvanic mode for simultaneous wastewater treatment and electricity generation. Figure 9 summarizes the different processes used for antibiotic removal with a stand-alone MFC [143]. MESs can enhance the removal rates of refractory organic pollutants through regulation of the microbial metabolism, co-metabolizing with the available matrix, and enhancing electron transport in the system [144]. Currently, researchers have primarily studied the electron transfer mechanism in bacterial systems, so the use of bacteria in MESs is more common [145].

Guo and colleagues found significant removal of chloramphenicol at lower chloramphenicol concentrations and high cathode potentials while at intermediate cathode potentials, both enrichment of chloramphenicol-resistant bacteria and the expression of ARGs were suppressed [146]. A study by Li et al. utilized a three-dimensional biofilm electrode reactor (activated sludge of wastewater treatment plant) to investigate the removal of sulfonazine, ciprofloxacin, and zinc from wastewater, and showed that the reactor not only had the ability to eliminate both antibiotics and zinc but also greatly reduced the risk of ARG transmission by removing the *intI*1 gene in wastewater [147]. Xue et al. reported that more than 85% of sulfamethoxazole was degraded within 60 h, which may have been due to the continuous electrical stimulation or the metabolism of microorganisms. In addition, sulfamethoxazole can be completely degraded into less harmful substances, such as alcohols and methane. *Shewanella* sp. and *Geobacteria* sp. have been shown to dominate this process in MFC power production while *Achromobacter*, *Alcaligenes*, and *Pseudomonas* also contributed to sulfamethoxazole degradation. It was also found that the production of ARGs by MESs was much lower than that in traditional sewage treatment plants [148]. Hua and colleagues applied MESs to degrade high concentrations of erythromycin. The removal rate reached 99% when the erythromycin concentration was 20 mg/L, and no ARGs were detected in the effluent [149]. These studies demonstrate that bioelectrochemical technology is not only an effective and reliable technique for treating antibiotic wastewater but also has great potential for reducing ARGs.

Furthermore, there is currently a gradual exploration of the use of extremophilic microorganisms in MES. Several studies have demonstrated biodegradation of the antibiotic chloramphenicol in biocathode MES at 10 °C, suggesting that antibiotic removal from the MES occurs after switching the temperature from 25 °C to 10 °C, despite a lowered reduction in chloramphenicol [150]. Clearly, the use of extremophiles provides an important way for bio-electro-chemical remediation of wastewater and biodegradation of antibiotics in extreme environments [151].

Although able to simultaneously generate electricity and treat wastewater, MESs have never been considered as a serious alternative in any of its promising application fields, chiefly due to their early stage development, high operating costs, and low performance efficiency [143].

### 3.6. Constructed Wetland

According to a study of 106 constructed wetlands (CWs) by Liu et al. [152], the removal efficiency of CWs for antibiotics showed good performance (averagely over 50%), especially vertical flow CWs (averagely up to 80.44%). Overall, the removal efficiencies of sulfonamide and macrolide antibiotics were lower than those of tetracycline and quinolone antibiotics. In addition, the relationship between the removal efficiency of antibiotics and COD, total suspended solids, total nitrogen (TN), total phosphorus (TP), and ammonia nitrogen (NH_3_-N) concentrations showed an inverted U-shaped curve with turning points of 300 mg/L, 57.4 mg/L, 40 mg/L, 3.2 mg/L, and 48 mg/L, respectively. The coexistence of antibiotics with nitrogen and phosphorus slightly reduced the removal efficiency of nitrogen and phosphorus in CWs. The removal effect of horizontal subsurface flow CWs for ARGs had better performance (over 50%) than that of vertical flow CWs, especially for sulfonamide resistance genes. Therefore, different types of CWs have different effects on the changes in physicochemical characteristics of sewage effluent. Compared with single mesocosm-scale CWs, Chen et al. reported that hybrid CWs between horizontal subsurface flow and vertical subsurface flow showed better performance in reducing the levels of COD, TN, TP, and NH_3_-N, with the removal rates of 40.9–76.6%, 23.9–89.5%, 60.1–93.0%, and 90.8–98.3%, respectively [153].

More important, it can effectively remove antibiotics in wastewater using matrix adsorption and interception, plant absorption and degradation, and microbial decomposition. Of these, microbial degradation and matrix adsorption play major roles in the removal process [154]. The microbial communities in CWs are better resistant to antibiotic stress, and it has been shown that microbial communities in CWs are more diverse than those in conventional activated sludge [155]. The activity and metabolic capacity of microbial communities in CWs were found to be significantly reduced when antibiotics were present in the inflow, although the activities recovered after 2–5 weeks [156].

In addition, the degradation of antibiotics by microorganisms in CWs is affected by conditions such as matrix type, plant species, dissolved oxygen, redox potential, temperature, and pH [157]. The matrix can not only adsorb antibiotics but also provides an attachment site for the microorganisms; thus, the antibiotics are preferentially adsorbed onto either the matrix directly or onto the biofilm coating the matrix. Adsorption onto the matrix increases the local concentrations of the antibiotics, promoting their degradation by the microorganism. The physicochemical properties of antibiotics play major roles in matrix adsorption, especially, water solubility and charge [158]. Hydrophobic antibiotics are more likely to be adsorbed than their hydrophilic counterparts [159]. While charged antibiotics are preferentially adsorbed onto the matrix via electrostatic interactions, neutral antibiotics are adsorbed through weak van der Waals interactions [160]. In addition to the physicochemical properties of antibiotics, adsorption is also affected by the matrix species, particle size, and other factors [152].

Wetland plants promote antibiotic removal in a number of ways. First, antibiotics are directly absorbed from wastewater by the roots. Second, plants provide oxygen and nutrients for microbial metabolism. Third, plants provide attachment sites for microorganisms [161]. A portion of the antibiotic absorbed by the roots of wetland plants is transferred to the stems and leaves by passive transport, while the remainder is transformed through glucosylation and glutindogenylation by plant enzymes [162]. Oxygen and material secreted by the plants are used by microorganisms within the rhizosphere, potentially enhancing the microbial degradation of antibiotics [163].

As an ecological sewage treatment technology, CWs have the advantages of low operating costs, good quality effluent, and easy maintenance. However, the treatment efficiency is affected strongly by weather and CWs also require large areas of land and extended treatment times.

### 3.7. Microalgae Technique

Microalgae can not only utilize a variety of nutrients in the wastewater (such as municipal, agricultural, and industrial wastewater) for their own growth but also effectively remove ammonia, nitrogen, phosphate, heavy metals, and organic pollutants from the wastewater at low cost [80]. The traditional strategies for improving pollutant removal by microalgae include the cultivation of mixed microalgae, domestication of microalgae in extreme environments, or the addition of common substrates such as acetate and formate to facilitate the degradation of contaminants [164]. The microalgae *Chlorella vulgaris* was cultured with 200 mg/L levofloxacin, leading to an increase in the removal rate of 1 mg/L levofloxacin by 16% after 11 days [165].

At the same time, it was found that microalgae can not only remove antibiotics through biodegradation, but their excreted photosensitive metabolites (such as extracellular polymeric substances) can remove antibiotics by indirect photolysis [166]. When exposed to light, the microalga *Chlorella* was found to produce ∙OH to induce the photolysis of norfloxacin [167]. Norvill and colleagues reported that the removal of tetracycline in a pilot-scale algal pool was mainly dependent on indirect photolysis during the day [168].

The mechanisms through which microalgae remove antibiotics can be summarized as follows and shown in Figure 10. First, the algal surface can strongly adsorb organic pollutants which can then be removed by photolysis [80,169,170]. Second, algae secrete enzymes that can accelerate the catalytic degradation of organic pollutants [169,170]. Third, algae increase the content of dissolved oxygen in water through photosynthetic oxygenation, increasing the ROS levels and thus promoting photolytic degradation of organic pollutants [169,170]. Fourth, algae and their secretions may act as photosensitizers to induce the indirect photolysis of organic pollutants [171]. However, studies on the removal of organic pollutants by microalgae are relatively limited and thus the predominant removal mechanism is unknown.

However, treatment with microalgae has several disadvantages and safety risks. First, microalgal photosynthesis may be inhibited by the toxicity of antibiotics and adverse wastewater conditions. Second, microalgal growth is dependent on the composition of the wastewater and the environmental conditions, increasing the complexity of regulating their use for antibiotic removal. Third, ARGs may accumulate and be transferred after the antibiotic treatment.

### 3.8. Hybrid Technology

Hybrid techniques have been developed for antibiotic removal. As shown in Figure 11, combinations between the use of microalgae, AOPs, CWs, MFC, activated sludge, membrane, and electrochemical methods have been investigated.

The combination of electrochemical methods and AOPs results in electrochemical AOPs (EAOPs) [172,173]. ∙OH can be produced directly in the oxidation reaction at the anode (electrooxidation/anodic oxidation), and can also be generated indirectly by Fenton’s reagent at the cathode (electro-Fenton). As shown in Figure 12, there is electrogeneration of ROS at the anode (Equations (6) and (7)), with subsequent oxidation of the organic molecules. In the presence of high Cl^−^ ion concentrations, reactive chlorine species (Cl_2_, HClO, and/or ClO^−^) are produced via reactions in Equations (8)–(10) along with ROS. Organic compounds can be effectively degraded by the reactive chlorine species with moderate oxidation [172,173].
M_(s)_ + H_2_O → M(HO∙) + H^+^ + e^−^(6)
M(HO∙) → M=O + H^+^ + e^−^(7)
2Cl^−^ → Cl_2(aq)_ + 2e^−^(8)
Cl_2(aq)_ + H_2_O → HClO + Cl^−^ + H^+^(9)
HClO ↔ ClO^−^ + H^+^(10)

At the cathode, there is an indirect in situ generation of ∙OH through the electro-Fenton reaction (Equation (11)) [172,173,174]. The electro-Fenton process can be upgraded by photochemical technology, leading to a more effective treatment of organics than with the photoelectric single catalytic effect due to photolysis of [Fe(OH)]^2+^ (Equation (12)), which accelerates the rate of organic mineralization. Moreover, photolysis of Fe(III) complexes can be induced by irradiation (Equation (13)), thus enhancing the efficiency of the process [172,173].
H_2_O_2_ + Fe^2+^ + H^+^ → HO∙ + H_2_O + Fe^3+^(11)
[Fe(OH)]^2+^ + hv → HO∙ + Fe^2+^(12)
[Fe(OOCR)]^2+^ + hv → R∙ + Fe^2+^ + CO_2_(13)

A combination of EAOPs and membrane filtration can effectively prevent the problems of membrane fouling (mainly by the in situ electrochemical process) and concentration polarization to varying degrees without backwashing [175]. Du et al. combined an integrated peroxymonosulfate-assisted electrolytic oxidation/coagulation with a ceramic ultrafiltration process to remove sulfamethazine [176]. The sulfamethazines were degraded by SO_4_^−^ and ·OH, generating large coagulated aggregates (206–275 µm) that significantly reduced the degree of membrane fouling. Tan et al. [177] employed multi-walled carbon nanotube-based electrochemical membranes in a mixed system to degrade antibiotics and found that the removal rates of amoxicillin, sulfamethoxazole, and ciprofloxacin were 98%, 95%, and 20%, respectively, and the removal rates of amoxicillin and sulfamethoxazole remained high even after four reuses.

Hybrids of CWs with MFCs have been intensively investigated in the past decade [178]. The bed of the CW comprises separate upper aerobic and lower anaerobic zones, which is similar to the aerobic and anaerobic chambers in MFC. This structural similarity between CW and MFC makes them compatible for merging, as shown in Figure 13. The merge of MFCs in CWs is conducive to the tuning of redox activities and electron flow balance in the aerobic and anaerobic zones in the CW bed matrix, which improves the availability of electron acceptors. The advantages of CWs–MFCs include high treatment efficiency, electricity generation, and refractory contaminant alleviation [178]. This novel hybrid technique has both features and addresses the limitations of the individual CW and MFC techniques [178]. Table 4 lists the studies on CWs–MFCs for antibiotic removal.

Furthermore, the integration of bioelectrochemical systems with CW–MFCs has also been investigated. Li et al. [189] combined a biofilm electrode reactor with MFC–CW (BER–MFC–CW), where the biofilm electrode reactor functioned as a pretreatment unit and the MFC–CW was used for further degradation. The removal rate of sulfamethoxazole in the biofilm electrode reactor unit was close to 90%, while the total removal rate in the combined system (BER–MFC–CW) was more than 99%. It was also found that the MFC–CW reduced the microbial community diversity and the ARG abundance in the biofilm electrode reactor. Zhang et al. constructed a BER–MFC–CW for sulfamethoxazole removal, observing a removal rate of over 99.29%, although the abundance of *sul* genes in the biofilm and effluent increased [190]. Zhang et al. reported that MFC–CW had no effect on the degradation products but enhanced sulfamethoxazole removal, and the methanogenic communities were found to be influenced by current [191].

Recently, hybrids between MFCs and AOPs have attracted attention [192,193,194]. For example, photo-assisted MFCs degrade not only organic electron donors at the anodes but also organic electron acceptors at the cathodes, together with electricity generation. In the presence of Fe^3+^, Fenton–MFCs can eliminate refractory organic substances by ·OH that is generated by Fenton reaction. Table 5 lists studies on the use of MFC–AOP hybrid systems for the removal of antibiotics. 

The combined use of microalgae and bacteria results in the formation of a unique phycosphere where the relationships between the two types of microorganisms range from cooperation to competition (Figure 14) [169]. Based on the cooperative interactions between microalgae and bacteria, a hybrid microalgae-activated sludge system is suitable for wastewater treatment for potential resource recovery [196,197,198]. Cephalosporin removal was investigated in a hybrid microalgae-activated sludge system, finding removal efficiencies up to 97.91% [77].

Previous studies have demonstrated that MFCs can effectively remove organic substances, although nutrient removal is limited due to anaerobic conditions at the anodic pole [199,200]. To solve these problems, microalgae have been introduced at the anode or cathode of MFCs; these microalgae perform oxidation reactions under anaerobic conditions at the anode and produce oxygen as electron acceptors at the cathode [201]. To date, a variety of microalgae have been reported to integrate with MFCs for wastewater treatment, mainly including the genera *Chlorella*, *Scenedesmus*, *Microcystis*, and *Chlamydomonas* [202,203].

Based on the cooperative interactions between microalgae and bacteria, hybrid microalgae–bacteria–MFC systems have been developed to deal with antibiotic pollutants. In these systems, cooperation between microbial metabolism and electrochemical redox reactions promotes the removal of antibiotics [199,204]. A novel microalgae–bacteria–MFC system was designed to simultaneously degrade anodic florfenicol, eliminate cathodic nitrogen, and generate bioelectricity [204]. Sun et al. [205] employed an algal–bacterial MFC to degrade florfenicol and ammonia. The results showed the complete removal of NH_3_-N within 90 h at the biocathode, while the degradation of florfenicol was enhanced through the anodic bioelectrochemical reaction; in addition, florfenicol promoted the growth of *Pseudomonas* species, leading to a 3.2-fold increase in power output. Furthermore, MFC systems can eliminate ARB and ARGs [200]. Therefore, a combination of MFC technology with a microalgal-based process can improve the overall performance, achieve sustainable wastewater treatment coupled with low greenhouse gas emission, and reduce sludge production [206,207]. The hybrid MFC–microalgae process is a novel alternative for wastewater treatment that also guarantees the sustainable recovery of renewable energy and biological products.

Despite high removal efficiencies and rapid removal rates [208], AOPs have high operational and maintenance costs and require large-scale applications, as well as the generation of large amounts of toxic byproducts [170]. To ensure efficient and economical treatment, studies have proposed combining AOPs with microalgal treatment, an alternative approach that has been demonstrated to be effective in the removal of antibiotics [140,209,210]. Table 6 shows the integration of microalgae with AOPs for antibiotic removal. 

## 4. Conclusions and Perspective

Antibiotics in natural environment can undergo photolysis, hydrolysis, and biodegradation, all of which are affected by various environmental factors, including pH, temperature, and the environmental medium. However, antibiotic removal by these methods is low and requires extended times for degradation. Thus, there is an urgency to develop more effective methods of antibiotic removal to reduce the contamination problem.

Biological methods have been traditionally used for the removal of pollutants. These have the advantages of low cost and large-scale operation. Unfortunately, antibiotics can inactivate the bacteria and increase the amounts of ARGs and ARBs, thus compounding the pollution problem. Adsorption technology and membrane separation technology have advantages of simplicity and convenience. In terms of adsorption technology, there is usually a significant gap between static adsorption effect under laboratory conditions and dynamic adsorption effect in practical engineering. In terms of membrane separation, the membrane is easily blocked and the application is also limited by the water quality conditions. AOPs can effectively remove most antibiotics with both high levels of efficiency and effective degradation. However, AOPs have different effects on different antibiotics and are susceptible to the complexity of antibiotic wastewater. The formation of intermediates during the reaction is highly likely. At present, we do not fully understand the toxicology and potential pathology of these intermediates. In addition, AOPs are also costly, with complex operational and maintenance issues.

Compared with the traditional biotechnology biotechnological applications, CWs and microalgal treatment have the advantages of effective antibiotic removal but nevertheless have the disadvantage of ARG transfer. MESs is not only an effective and reliable technique for treating antibiotic-containing wastewater but also has great potential for the reduction in ARGs. The relatively low concentrations of antibiotics in the environment present difficulties for their removal by traditional treatment techniques. It is hard to achieve ideal removal by relying solely on one technique. This suggests that the advantages of different treatment techniques should be combined for the effective removal of antibiotics. Many studies on hybrid techniques, combining EAOPs, CW–MFC, microalgae-activated sludge, and AOPs–MFC, have demonstrated their effectiveness.

Research into the different techniques for antibiotic removal has been fruitful, although most are currently at the laboratory stage. This indicates that there is a significant gap between the transitions from the laboratory environment to the practical application. It is important to apply these findings as soon as possible and, in particular, the application of hybrid technology to achieve effective removal in large-scale operations should be the focus of future research.

## Figures and Tables

**Figure 1 ijerph-19-10919-f001:**
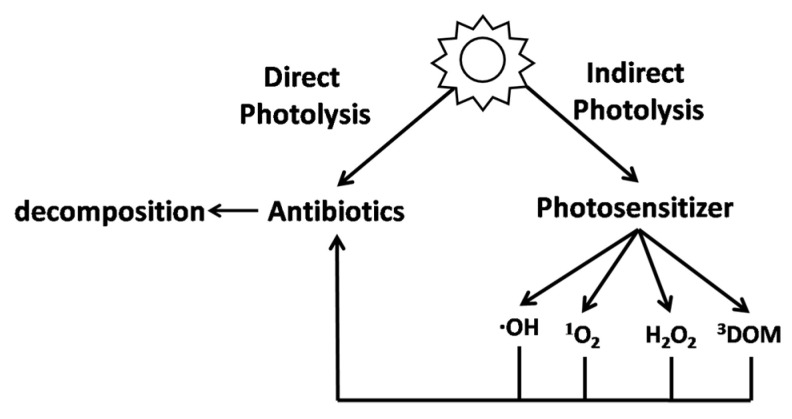
Comparison of direct and indirect photolysis (DOM: dissolved organic matter). Reproduced with permission from Arnold and McNeill, Analysis, Fate and Removal of Pharmaceuticals in the Water Cycle [53]; published by Elsevier, 2007.

**Figure 2 ijerph-19-10919-f002:**
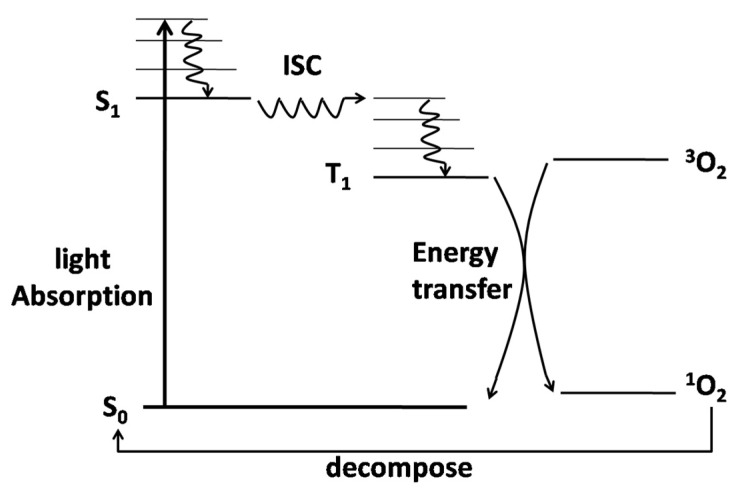
Self-sensitized photolysis by the photosensitized excitation of oxygen in its ground triplet state (^3^O_2_) to its lowest singlet excited state (^1^O_2_) (S_0_: photosensitizer ground state; S_1_: singlet excited state; ISC: intersystem crossing; T_1_: triplet state). Reproduced with permission from Martin et al., *J. Photochem. Photobiol*. [63]; published by Elsevier, 2017.

**Figure 3 ijerph-19-10919-f003:**
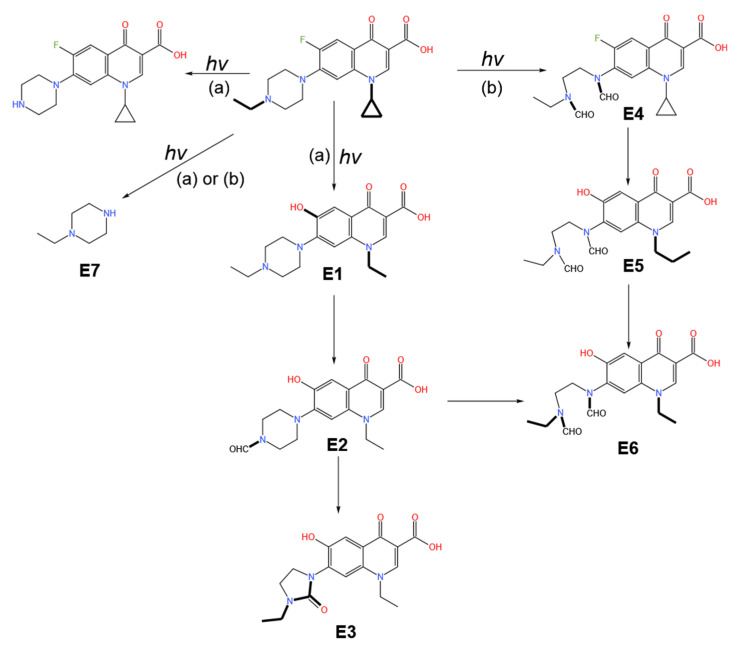
Photolysis pathways of enrofloxacin at pH4 (**a**) and pH8 (**b**). Reproduced with permission from Babic et al., *Chemosphere* [68]; published by Elsevier, 2013.

**Figure 4 ijerph-19-10919-f004:**
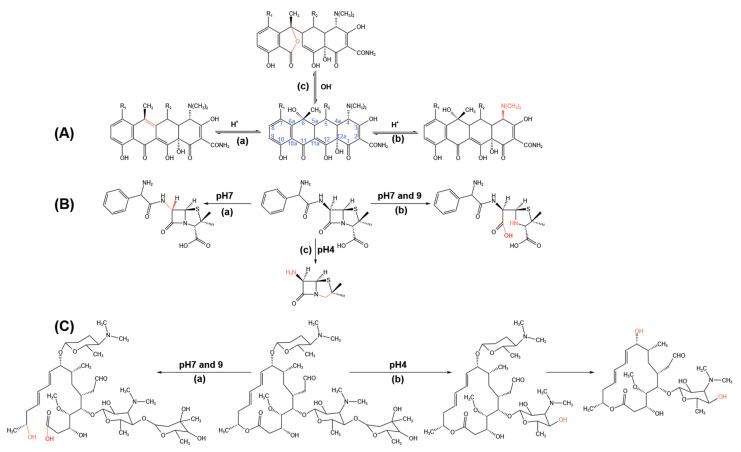
Proposed hydrolysis pathways for (**A**) tetracycline, (**B**) ampicillin, and (**C**) spiramycin according to [71,72,73,74,75].

**Figure 5 ijerph-19-10919-f005:**
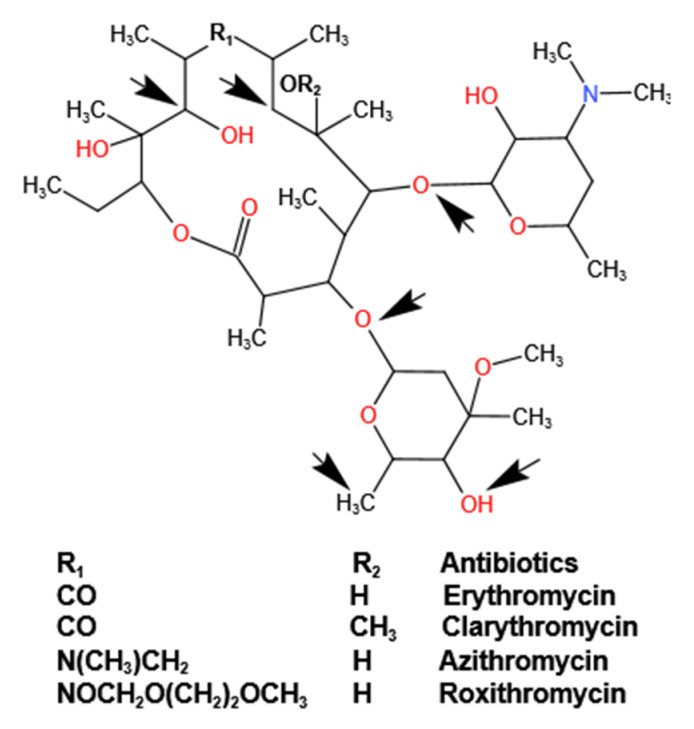
The possible attack site of ∙OH against several macrolides. Reproduced with permission from Babic et al., *Curr. Opin. Green Sust*. [121]; published by Elsevier, 2017.

**Figure 6 ijerph-19-10919-f006:**
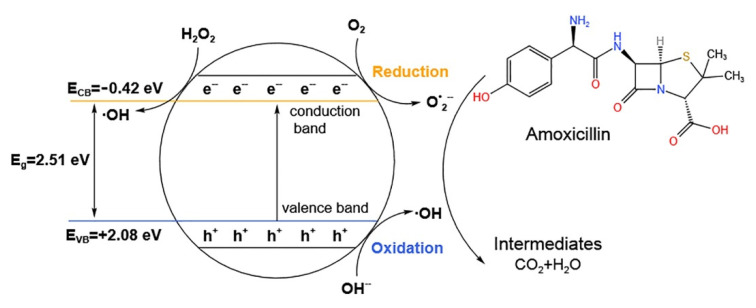
Schematic diagram of the degradation mechanism of the antibiotic amoxicillin over the modified Ce_0.04_Sr_0.96_Fe_0.04_Ti_0.96_O_3_ photocatalyst. Reproduced with permission from Suwannaruang et al., *Surf. Interfaces* [123]; published by Elsevier, 2021.

**Figure 7 ijerph-19-10919-f007:**
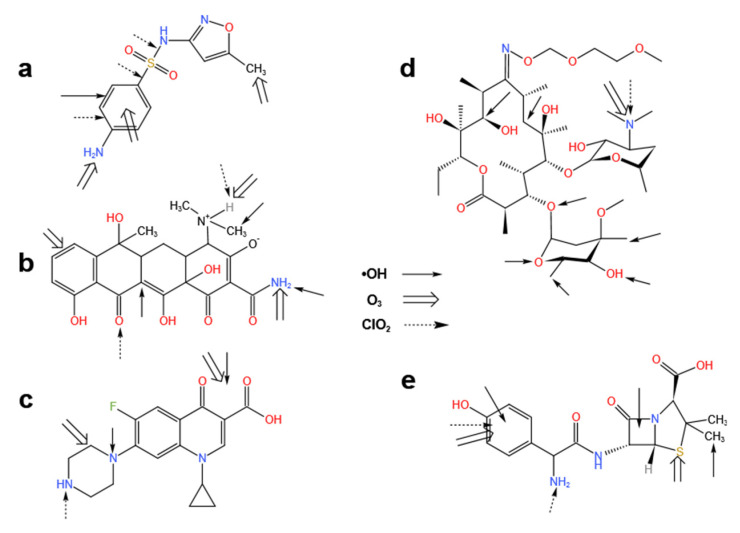
The attack sites of O_3_, ∙OH, and ClO_2_ against five antibiotics ((**a**) sulfamethoxazole, (**b**) tetracycline, (**c**) ciprofloxacin, (**d**) roxithromycin, (**e**) amoxicillin) according to [124,125,126,127,128,129,130,131,132,133,134].

**Figure 8 ijerph-19-10919-f008:**
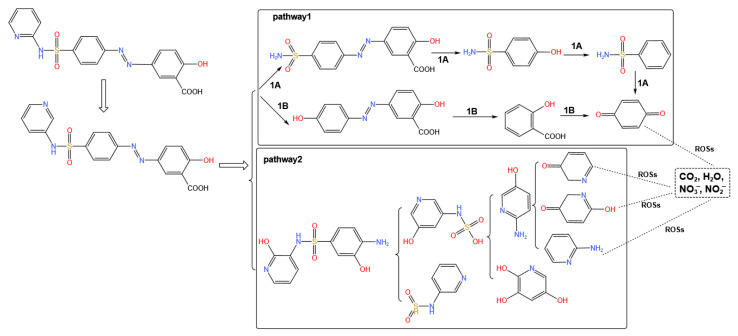
Reaction pathway of oxidation degradation of sulfasalazine by O_3_/H_2_O_2_ according to Pelalak et al. [138].

**Figure 9 ijerph-19-10919-f009:**
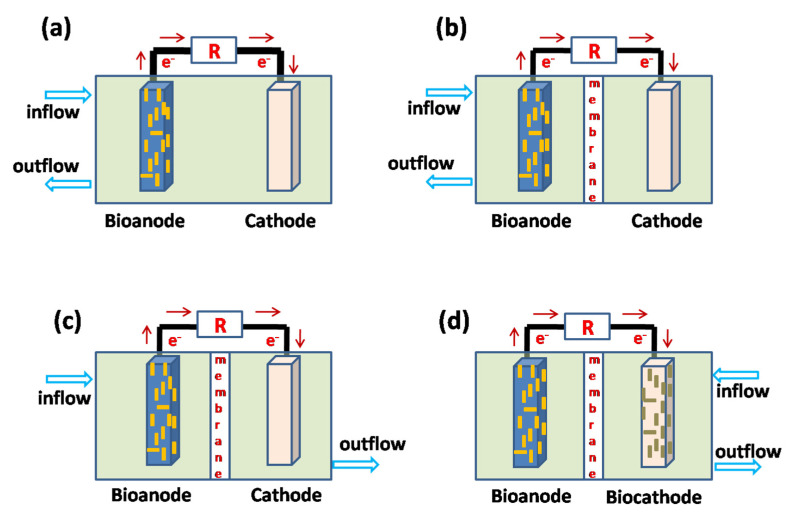
Different processes for antibiotics removal in MFCs: (**a**) single-chamber MFC, (**b**) dual-chamber MFC, (**c**) sequential anode-cathode operation in a dual-chamber MFC, and (**d**) biocathode operation in a dual-chamber MFC according to Zakaria and Dhar [143].

**Figure 10 ijerph-19-10919-f010:**
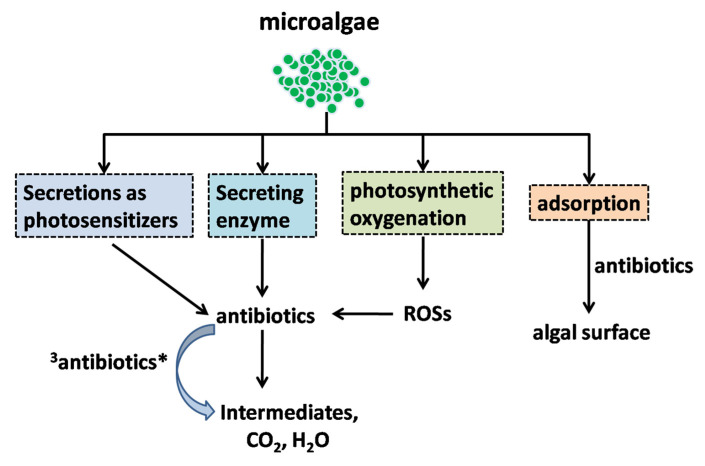
Putative antibiotic removal mechanisms by microalgae (^3^antibiotics*: triplet excited-state antibiotics).

**Figure 11 ijerph-19-10919-f011:**
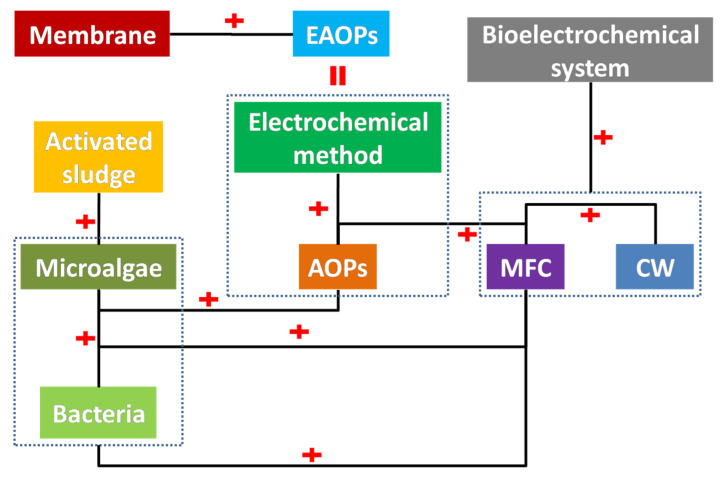
Hybrids of different techniques for antibiotic removal.

**Figure 12 ijerph-19-10919-f012:**
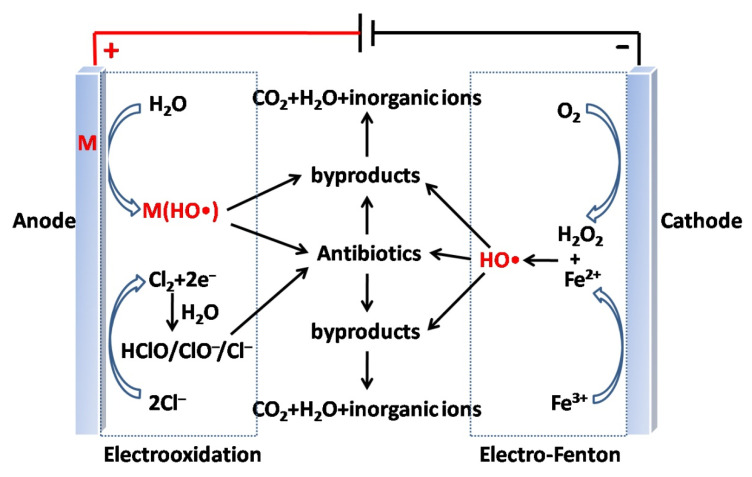
Schematic representation of the mechanism of oxidant production in electrooxidation and electro-Fenton processes (M = Fe, Mn, Ni, Co, Nb, Ti, Zn). Reproduced with permission from Ganiyu et al., *Appl. Catal. B: Environ*. [172]; published by Elsevier, 2020, and Ganiyu et al., *Appl. Catal. B: Environ*. [173]; published by Elsevier, 2018.

**Figure 13 ijerph-19-10919-f013:**
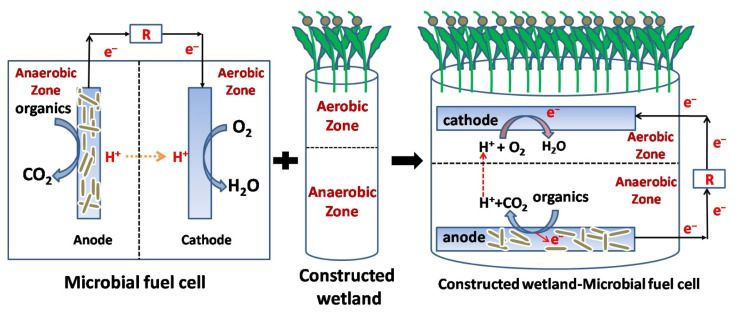
The similarities between microbial fuel cell (MFC) and constructed wetland (CW) configurations. Reproduced with permission from Ganiyu et al., *Bioresour. Technol*. [178]; published by Elsevier, 2021.

**Figure 14 ijerph-19-10919-f014:**
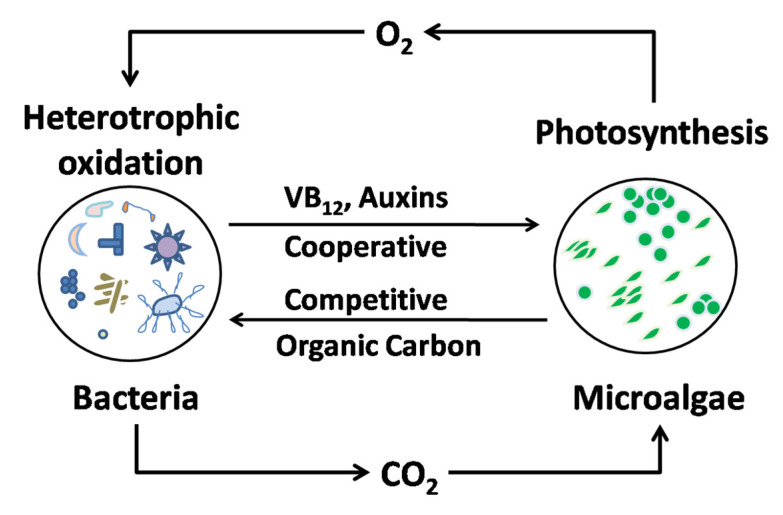
Cooperation and competition relationships between microalgae and bacteria (VB_12_: vitamin B_12_). Reproduced with permission from Xiong et al., *Environ. Int.* [169]; published by Elsevier, 2021.

**Table 1 ijerph-19-10919-t001:** ARBs and ARGs detected in the natural environment.

Antibiotics	Site	ARGs	ARB
Tetracyclines, oxytetracycline	Surface water, China [7]	*bla_ctx_*, *sul* I, *tet* A, *qnr* S, *aac*-1b and so on	antibiotic-resistant *Escherichia coli*
Sulfonamides, sulfonamides	River, China [4]	*sul* 1, *sul* 2, *tet* A,*tet* B,*tet* E,*tet* W,*tet* M,*tet* Z	River bacteria
Trimethoprim, Ofloxacin	River, wastewater treatment plant, Spain [8]	*qnr* S, *erm* B, *tet* W, *bla*TEM, *bla_NDM_*, *bla_kpc_*, *van* A	
Tetracyclines, Chloramphenicol	River, Germany and Australia [9]	*Ampc*, *van* A, *tet* A, *aac(3)*-IIa, *dfr*A1, *erm*A	
β-lactams	River, United States [10]		antibiotic-resistant Gram-negative bacteria
Tetracyclines, amoxicillin	River, France [11]		antibiotic-resistant *Escherichia coli*

**Table 2 ijerph-19-10919-t002:** The half-life times of antibiotics reported in different environment.

Antibiotics	Total	Photolysis	Biodegradation
	Half-Life Time (t_1/2_, d)	Half-Life Time (t_1/2_, d)	Half-Life Time (t_1/2_, d)
Sulfadiazine	25.7 (stream) [45]		
Sulfamethazine	17.3 (stream) [45] 12.9 (lakewater) [46]	3.4 (h) [47] 34.7 (lakewater) [46]	24 (freshwater) [47] 7.79 (seawater) [47]
Sulfamethoxazole	17.8 (stream) [45] 15.5 (h, river) [48] 34 (hyporheic zone) [49] 11.4 (lakewater) [46]	3.73 (h) [47] 49.8 (lakewater) [46]	13 (seawater) [47]
Sulfadimethoxine	18.2 (stream) [45] 10.5 (lakewater) [46]	31.9 (lakewater) [46]	
Sulfamerazine	17.9 (stream) [45]		
Sulfathiazole	13.3 (stream) [45]		
Enrofloxacin	8.78 (stream) [45]		
Norfloxacin	5.64 (stream) [45]	0.03(h) [47]	15.1 (freshwater) [47] 8 (seawater) [47]
Novobiocin		1.6 (h) [47]	8.25 (freshwater) [47] 2.88 (seawater) [47]
Ofloxacin	11.1 (stream) [45]		
Ciprofloxacin	5.33 (stream) [45]	0.04 (h) [47]	15.75 (freshwater) [47] 8.3 (seawater) [47]
Tetracycline	4.15 (stream) [45]		
Oxytetracycline	1.82 (stream) [45]		
Erythromycin	4.22 (stream) [45]		
Roxithromycin	2.76 (stream) [45]		
Fluoroquinolone		18.4 (river) [50]	10.4 (river) [50]
Enrofloxacin		0.8, 3.7, 72 [51]	
Flumequine		3.21 (h) [47]	29.2 (freshwater) [47]
Sulfamethoxypyridazine		1.58 (h) [47]	14 (freshwater) [47] 11.25 (seawater) [47]
Trimethoprim			6.63 (seawater) [47]

**Table 3 ijerph-19-10919-t003:** The antibiotic concentrations in the inflow and outflow of wastewater treatment plants (ng/L).

Antibiotics	Inflow Concentration	Outflow Concentration	Site
Cefalexin	670–2900	240–1800	Hongkong, China [95]
Cefotaxim	24	34	
Sulfadiazine	120–320	120–230	
Tetracycline	96–1300	180–620	
Norfloxacin	110–460	85–320	
Cefalexin	13818	5624	United States [96]
Tetracycline	240–48,000	50–3600	
Sulfamethoxazole	650–4255	86–4145	
Ciprofloxacin	315–5451	130–919	
Sulfamethoxazole	246	46	Italy [97]
Amoxicillin	18	-	
Ofloxacin	463	191	
Clarithromycin	319	117	
Lincolnensin	9	6	
Chloramphenicol	4–452	6–69	UK [98]
Trimethoprim	464–6796	685–3052	
Erythromycin-H_2_O	7,110,025	232,841	
Sulfapyridine	2164–12,397	94–1112	
Sulfamethoxazole	3–274	3–44	
Sulfamethoxazole	2.1–809	506	Harbin, China [99]
Azithromycin	110	61.2	
Clarithromycin	321	164	
Roxithromycin	14.2–2986	6–1419	
Ofloxacin	1.5–2787	1.1–1481	
Norfloxacin	1.5–2168	0.8–1018	

**Table 4 ijerph-19-10919-t004:** Hybrids of constructed wetlands coupled with microbial fuel cells (CWs–MFCs) for antibiotic removal.

Antibiotic (Initial Concentration)	Removal Efficiency	Key Findings
Sulfadiazine (4 mg/L) [179]	-	- Sulfadiazine removal and ARGs abundance were higher in the closed-circuit operation than the open circuit; - Low hydraulic retention time (HRT) led to higher sulfadiazine accumulation and ARGs abundance on the electrode; - Both adsorption and biodegradation contributed to antibiotic removal.
Sulfamethoxazole (5–100 µg/L), tetracycline (5–50 µg/L) [180]	99.70–100% (sulfamethoxazole), 99.66–99.85% (tetracycline) at HRT of 1 day	Plants and circuit connection both accelerated antibiotic removal.
Sulfadiazine (2 mg/L) ciprofloxacin (2 mg/L) [181]	-	A low level of Zn enriched ARGs, while excessive Zn inhibited antibiotic removal and ARGs proliferation.
Sulfadiazine (2 mg/L) and ciprofloxacin (2 mg/L) [182]	Averagely, 80% for sulfadiazine and 90% for ciprofloxacin	Methane emission declined by 15.29% in CW–MFC compared with CW.
Tetracycline and sulfamethoxazole (400–1600 µg/L) [183]	More than 99% for tetracycline and sulfamethoxazole	- ARGs abundances significantly affected by effluent antibiotics; - Power density decreased when antibiotic concentration increased from 400 to 1600 µg/L.
Sulfamethoxazole (4 mg/L) [184]	Mean 82.37%	- Removal efficiencies of total nitrogen, ammonia nitrogen, and sulfamethoxazole were higher for CW–MFC than CW; - Copy number of sulfamethoxazole ARGs was much lower for CW–MFC than CW.
Sulfamethoxazole (100 µg/L), tetracycline (50 µg/L) [185]	96.9–98.2% for sulfamethoxazole and 80.3–88.0% for tetracycline	Sponge iron (s-Fe^0^) significantly reduced ARGs and improved voltage output, power density, columbic efficiency, and reduced internal resistance of reactor.
Sulfamethoxazole (60 µg/L), tetracycline (25 µg/L) [186]	88.24–99.4% for sulfamethoxazole, 84.6–97.8% for tetracycline	carbon source species and concentrations, external resistances, and aeration duration all play vital roles in removing sulfamethoxazole and tetracycline.
Sulfadiazine (2 mg/L), ciprofloxacin (2 mg/L) [187]	84.9–95.9% in graphite CW–MFC and 46.6–62.8% in Mn ore CW–MFC for sulfadiazine, >97.8% for ciprofloxacin	- Better COD removal and higher bacterial community diversity and electricity generation performance in Mn ore CW–MFC; - lower concentration of sulfadiazine and ARGs abundances in effluent of graphite CW–MFC due to higher graphite adsorption and filter capacity.
Sulfonamide (4 mg/L) [188]	-	Closed circuit operation with low HRT enhanced sulfonamide mass accumulation on electrodes by electro-adsorption, and thus higher ARGs abundance was detected in electrodes and effluent.

**Table 5 ijerph-19-10919-t005:** Studies on the antibiotic removal by the hybrids between MFCs and AOPs.

Integration of MFC with AOPs	Features
Bio-Electron-Fenton (BEF)+MFC [192]	- γ-FeOOH graphene as cathode; - After 40 h treatment, degradation rate of sulfamethoxazole and norfloxacin were 97.4% and 96.1%, respectively; - Compared with the sludge digestion system, residual in sludge declined from 10.2 to 1.1 and from 31.3 to 3.1 for sulfamethoxazole and norfloxacin, respectively.
Photo + Fenton + MFCs [193]	- Mo and W immobilized onto graphite felt cathode; - Under aerobic conditions, photogenerated electrons favored O_2_•^−^ production over •OH, in the presence of Fe(III), •OH predominant over O_2_•^−^; - Under anaerobic conditions, photogenerated holes directly involved into metronidazole oxidation.
Photo + MFCs [194]	- Polyaniline-carbon nanotube/stainless steel (PANi@CNTs/SS) bioanode and LiNbO_3_/carbon felt (CF) photocatalytic cathode; - Maximum open circuit potential and power output of 0.806 V and 0.546 W/m^2^, with calculated internal resistance of 340 Ω; - Ofloxacin removal efficiencies were 86.5%, 81.2%, 76.1%, 70.2% at0.1, 0.2, 0.3, and 0.4 mmol/L.
MFC + Fenton [195]	- Graphite rod with stacked graphite granules used as the electrode; - Generally, closed circuit showed significantly higher capacity to degrade sulfamethazine and triclocarban than open circuit in both the batch mode and the continuous flow.

**Table 6 ijerph-19-10919-t006:** Integration of microalgae with AOPs for antibiotic removal.

Integration of Microalgae with AOPs	Antibiotic Removal Efficiency	Key Findings
Fenton + microalgae [209]	Amoxicillin (96.86–99.86% at 100, 200 and 300 mg/L, less than 90% at 400 and 500 mg/L) cefradine (93.98–95.5%)	Compared with removal capacity of individual algal treatment, a higher removal rate and a shorter treatment time were achieved using combined Fenton–algal treatment
UV + microalgae [140,210]	Cefradine, 73% (UV treatment), 78% (UV–algae treatment)	- UV treatment increased the effluent toxicity (1.04 times of the parent compound) - UV–algae combined treatment reduced the effluent toxicity (nearly half of that by UV treatment)
	84.96–99.84% for amoxicillin, 44.42–63.18% for cefradine	Optimal application involved UV at 365 nm combined green algae *Scenedesmus obliquus*
Fe(III) + microalgae [211]	Enrofloxacin (about 80–50% at 1–9 mg/L) and ciprofloxacin (about 80–40% at 10–80 mg/L)	Degradation efficiency of enrofloxacin and ciprofloxacin was better at lower concentrations

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
