# Peer review of "Current Progress in Natural Degradation and Enhanced Removal Techniques of Antibiotics in the Environment: A Review"

_ijerph, 2022, doi:10.3390/ijerph191710919_

Round 1

Reviewer 1 Report (Previous Reviewer 1)

The revised manuscript was significantly improved, namely by the introduction of figures that considerably help on the understanding of the processes associated to the antibiotic’s removal techniques. A good revision of the subject was done well supported by references, although no references from 2022 were included and should be. However, the English still needs revision. In fact, several parts of the manuscript are still difficult to understand. Hence, the revision of the manuscript by an English native speaker should be mandatory, otherwise some parts of the text are difficult to follow and to understand the meaning.

Abstract

Lines 18-19

What the authors mean with the following sentence?

“There is a big gap for adsorption and membrane to apply in the practice.”

Lines 24-25

What is the meaning of EAOPs. The first time an abbreviation appears its meaning has to e indicated.

Introduction

Table 1

It should be Spain and not Spanish

Line 121-123

I still not understand the following sentence: “Besides, when antibiotics enter the water, they may be enriched by suspended solids, microorganisms, algae and microplastics in the water [38].

2. The degradation behaviors in the natural environment

The authors mention microbial and plant degradation. Why was algae biodegradation not included? Do algae just act through adsorption?

Line 152

Light energy transformation? Or light energy transference? In fact all the paragraph requires revision.

2.3 Biodegradation

The sentence that was changed still needs revision such as “The antibiotic structures can be damaged and become inactivated by the intracellular or extracellular enzymes of ARB, and the mechanism of which could be hydrolysis, group transfer (such as acetyl transfer), and redox [42]”.

The last part of the sentence (in red) is not understandable. What is the relation between that part of the sentence and the first part? Are you mentioning the mechanisms by which inactivation of the enzymes may occur? It is not clear.

Figure 1. In the caption of Figure 1 the authors mention photosensitizer species such as NO3- and NO2-. However, in the figure itself those are not the Photosensitizer species mentioned. Thus, the caption is not consistent with the figure. Other photosensitizer species such as NO3- and NO2- can be mentioned in the text, but not in this caption.

Figure 3 The photolysis pathways of enrofloxacin at pH=4 (a) and pH=8 (b)…….

Reaction 6

This reaction is not balanced.

Figure 13

…..according to Xiong et al……

4. Conclusion and perspectives

Line 734

Simplicity not simple

Author Response

Reviewer 2 Report (New Reviewer)

The manuscript deals with a very important problem, which is the removal of antibiotics, which are important pollutants in environmental systems. Therefore, the manuscript deals with a topic of global interest and highly publishable. Before being published, authors must make the following changes.

1.     Keywords: Do not just repeat the title. Please rearrange.

2.     Introduction: The lines 92-94 should be explained better how antibiotics enter in groundwater.

3.     The degradation behaviors in the natural environment: (1) The half-lives of the main antibiotics should appear in this section. (2) Hydrolysis. More examples of hydrolysis should appear. (3) Biodegradation. Please add to previous research about degradation by enzymes in plant roots (lines 288 and 289).

4.     The enhanced removal techniques: (1) Why might the effluent concentration of the antibiotic cefotaxime be higher than the influent? (Table 2) (2) In lines 400-401 and 442 separate the words, the spaces are missing.

Author Response

Reviewer 3 Report (New Reviewer)

The document is a revision on the fate of antibiotics in the environment, and the different treatments that can be used to remove these emerging contaminants.

The document, which I assume is a revised version, is well organized, has interesting information, and can definitively contribute to the current debate on the increase of ARB and ARG in the environment. 

Author Response

Dear the Reviewer,

   Thank you very much for the positive comments. And thanks again for your valuable comments on our manuscript.

Best wishes,

Fengxia Yang & Yongzhen Ding

This manuscript is a resubmission of an earlier submission. The following is a list of the peer review reports and author responses from that submission.

Round 1

Reviewer 1 Report

General comments

This revision presents a collection of information from different sources, sometimes confused, not always scientifically accurate and with lack of criticism. The organization of the manuscript may also be improved namely by the inclusion of additional figures illustrating the different possible mechanisms by which antibiotics can be removed /detoxicated.

I recognize the difficulty of the thematic that involves several areas of knowledge, but clearly this manuscript needs to be improved before being in conditions to be submitted.

In general, the manuscript is difficult to read, and some parts are even hard to understand and follow since many sentences are not written in a clear English thus, requiring a language revision.

Some specific comments are indicated below aiming the authors to prepare a new and improved version adequate for submission.

Abstract

Lines 21 and 22

Adsorption and membrane separation are convenient and no intermediates are formed, but they do not really degrade antibiotics.

Line 26

“Considered the complexity of antibiotics pollution,…”

Introduction

Line 38

“…..caused by pathogenic bacteria.”

Line 71-73

The following sentence is not clear and requires revision:

“When antibiotics enter the water, the suspended solids, microorganisms, algae, and microplastics in the water are likely to continuously enrich various antibiotics [19].”

Line 73

“Based on the measured toxic data of antibiotics,….” Too vague, please specify which measured toxic data.

2. The degradation behaviors in the natural environment

Lines 88-89 “…. and the biodegradation includes the microbial degradation and the plant degradation.” This sentence should be better clarified.

Figure 1 The caption should also contain the reference from which that info was obtained.

2.3 Biodegradation

Lines 171-172

The following sentence requires revision:

“The antibiotic structures can be damaged and become inactivation by their intracellular or extracellular enzymes…….”

Lines 178 and 179

According to the authors: Maki and colleagues domesticated and selected effective strains with efficient degradation…….”

What the authors mean by domesticated strains? How was that done?

Lines 183-184

The authors state that the advantages of microbial degradation to treat antibiotic pollutants include no secondary pollution. Please explain what is secondary pollution.

Please note that sometimes the biodegradation products can even be more harmful than the original compounds and that is why it is very important to monitor not only the initial compound but also the formation of possible metabolites.

Lines 185-186

According to the authors:

“Plant adsorption and degradation refers to the degradation, transformation, absorption, metabolism, and detoxification of pollutants by plants or their root secretions to repair the contaminated soil, water, and atmospheric environment [39].”

Adsorption and degradation by plants or other organisms can be independent processes, thus the sentence is not clear and accurate.

Reviewer 2 Report

Manuscript ID: ijerph-1735815

Type of manuscript: Review

Title: Current Progress in Natural Degradation and Enhanced Removal Techniques of Antibiotics in the Environment: A Review

Journal: International Journal of Environmental Research and Public Health

Comment 1: Table 1 – Clear the relationship between protein kinase inhibitors and antibiotic resistance (with full consideration to the role of enterobacteria in the explanation of the mechanism) and explain how to tackle antimicrobial resistance (consider the zoonosis as an emerging link to antibiotic resistance) and antibiotic consumption monitoring considering the potential causes and strategies for prevention. Could you assess the affected bird communities and unicellular organisms with the spread out of antibiotics in nature? (Consider the interactions of antibiotics with different host factors in these environments)

Comment 2: "Table 2 shows the antibiotics concentrations in the inflow and outflow of some wastewater treatment plants. Seriously, the levels of ARB and especially ARGs in urban sewage were significantly high when compared with the natural or less affected water bodies" 1- Is the treated wastewater safe to reuse for agricultural irrigation? 2- Discuss the potential impacts of disinfection processes on the elimination and deactivation of ARG during treatment.

Note 1: "Therefore, the persistent residue of antibiotics in the environment has brought challenges to the structural stability of ecosystem and human health" 1- Does human activity impact the natural antibiotic resistance background? 2- Consider the ranking risk in resistomes. 3- Clear the risk of antibiotic resistance bacteria (ARB) and antibiotics resistance genes (ARGs) to the environment and public health via human health risk assessment (HHRA).

Comment 3: "Kumara et al. found that corn (Zea mays L.), green onion (Allium cepa L.), and cabbage (Brassica oleracea L. Capitata group) could absorb chlortetracycline in soil, but had limited absorption and removal on tylosin" Study the extent of resistant organisms and horizontal spread of ARB among bacteria from human settlements into the environment with details about antibiotics and AMR/ARGs in soil.

Note 2: Provide the results of metagenomic analysis for ARGs and their genetic compartments in the environment.

Note 3: "The antibiotic structures can be damaged and become inactivation by their intracellular or extracellular enzymes, the mechanism of which could be hydrolysis, group transfer (such as acetyl transfer), and redox" Explain the role of horizontal gene transfer (incl. inter-species population dynamics) in the spread of extracellular ARGs in soil.

Comment 4: "The seasonal changes were shown in the species and concentration of nineteen antibiotics from twenty-seven groundwater samples" Explain the seasonal variation of antibiotics concentration in different environments.

Note 4: Define the links to seasonal variations and the driving factors of antibiotic resistome in aquatic environment.

Comment 5: "Antibiotics exist wildly in the environment and can be all detected in surface water, groundwater, sediments, soil, and even drinking water [11][12], indicating that the pollution of antibiotics is widespread" Illuminate the speciation, occurrence, and fate of antibiotics in marine environment and link some of your findings to the leaching and hydrochemistry of the nature of water.

Note 5: "Liang and colleagues used the combination of ultrafiltration and two-stage reverse osmosis to treat piggery wastewater, which not only removed pollutants such as nitrogen and phosphorus, but also trapped 72.64% of ARGs, effectively reducing the risk of ARGs to the natural water bodies" and "The study of Li and colleagues used a 3-dimensional biofilm electrode reactor (activated sludge of wastewater treatment plant) to treat sulfonazine, ciprofloxacin, and zinc in wastewater, and showed that the reactor not only had the ability to eliminate both antibiotics and zinc, but also greatly reduced the risk of ARGs transmission" 1- More information about risk assessment are requested. 2- Mention the depths factor during sampling.

Comment 6: In relation to Ref.[41]: Kumara et al. (2005), could you clarify the influence of manure application on antibiotic distribution and ARGs extents in groundwater and surface water?

Note 6: The interactions including exchange between surface water, groundwater, and terrestrial ecosystems (consider the varied characterization of these water systems) and their effects on natural degradation and antibiotics concentrations in environment must be explained. (Involve in your response the movement of water through groundwater and the available models and hydraulic and geochemical methods used)

Note 7: "Draining from soil, infiltration from sediment, and introduction through the exchange between surface water and groundwater are three main sources of antibiotics in groundwater" More data about nutrients exchange and water isotopic signatures effect on the mechanism of antibiotics infiltration and drainage from soil must be provided and explained.

Comment 7: "Based on the measured toxic data of antibiotics, more than half of antibiotics had toxic effects on fish, and nearly 20%, 33%, and 44% of antibiotics showed strong toxic effects on algae, fish, and Daphnias, respectively" and "Maki and colleagues domesticated and selected effective strains with efficient degradation and strong tolerance from the sediment of cultured seawater fish, which had significant degradation functions against oxytetracycline and doxycycline" Please, consider the kinds of antibiotic profiles when describing the main sources of antibiotics in algae, fish, and Daphnias.

Comment 8: "Strong photosensitizers can accelerate the photolysis of antibiotics by capturing free radicals (such as ·O2- and ·OH)" Consider the associated hydrogen evolution from water when using these photosensitizers.

Consider 9: "The photosensitizers are widely found in nature, like humus, riboflavin, N3-, NO2-, Fe3+, Fe2+, NaCl, and so on." Consider the Fe3+/Fe2+ interconversion and the conditions for the formation of NOCl.

Comment 10: "Studies have shown that antibiotics, like fluoroquinolones, tetracyclines and chloramphenicol, undergo not only direct photolysis, but also self-sensitized photolysis" What are the effects of natural organic matter (OM) and nitrate to transformation product formation? Define the wavelengths of the UV-direct photolysis of antibiotics (consider their properties and study the impacts of perfluoroaldehydes and fluorotelomer aldehydes) and study the reaction kinetics and the formation of the equally potent of them.

Comment 11: "Through the anaerobic-aerobic combined biological method to treatment the piggery wastewater (mainly including sulfonamides and β-lactam), it was reported that the anaerobic digestion process mainly reduced the chemical oxygen demand (COD), the aerobic biological process had a significant contribution to the removal of antibiotics, and in three days, the total removal rates of COD and antibiotics were 95% and 92%, respectively" Define the working and environmental conditions and factors and consider in your explanation the degradation, transformation products, and antibacterial activity.

Comment 12: Table 2, "Chen and colleagues found that under aerobic and anaerobic conditions, the removal rates of nine antibiotics (sulfonamide monomexazine, sulfamidazine, sulfamadazine, trimethoprim, norfloxacin, ofloxacin, lincomycin, leucomycin, oxytomycin) in piggery wastewater by aeration biofiltration system were all greater than 82%", and "During light, microalgae Chlorella were found to produce hydroxyl radicals to induce the photolysis of norfloxacin" 1- Is it feasible to use a photodegradation-based treatment for the removal of antibiotics? 2- The photocatalytic and oxidative degradation of these antibiotics must be revealed and explaind (incl. efficiencies, photochemical degradation kinetics and mechanisms). 3- The degradation and transformation processes must be elucidated using new illustrations.

Comment 13: "Iron can exist in the form of Fe(OH)2+ in aqueous solution, and are converted to Fe3+ through a series of reactions under light with the productions of ∙OH and HO2∙, which promote the photolysis of antibiotics" Study the spatial heterogeneity of indoor OH and HO2 and explain the mechanism of Fe(Oh)2+(Aq) photolysis in aqueous solution.

Note 8:            1- Test the hydroxyl radical formation upon photolysis. 2- Measure the heats of ionization of HO2 and OH in aqueous solution.

Comment 14: "…which can spread to the organisms and the humans through horizontal gene transfer routes in the direct contact or the food chain…" You must must describe in brief the analytical and physicochemical methods used for identifying antibiotic residues in different matrixes including foods (as honey) and MSW and discuss the relative challenges in analytical chemistry.

Comment 15: "Auromycin, tetracycline, β-lactaides, and macrolides are found to be prone to hydrolysis, while quinolones and sulfonamides are found to be uneasy to hydrolysis" Clarify the mechanisms' actions of these compounds and their resistances.

Comment 16: "The removal mechanism is mainly based on the "in situ" generation of hydroxyl radicals, which react rapidly with most organic compounds (except chlorinated alkanes), but lack attack-selective" 1- Explain these reactions and the associated mechanisms, especially with presence of NOx, nitrate radicals, ozone, hydrogen bond acceptors, chlorinated VOCs, and semi-VOCs. 2- Refer to conditions of the increased hydroxyl radical generation and explain the contribution to the oxidation mechanism of ferrous ions by hydroxyl radicals in the presence of organic compounds (the rates, reactivity, and mechanism for reaction of organic compounds with hydroxyl radicals must be mentioned in your response).

Comment 17: "Many AOPs are based on a combination of strong oxidants (e. g. ozone or hydrogen peroxide), with catalysts (e. g. transition metal ions or photocatalysts) or radiation (e. g. ultraviolet or ultrasound). However, many other oxidants are selective oxidants and only react with some of the groups in the molecular structure of antibiotics. The attack sites of O3, ∙OH, and ClO2 against sulfamethoxazole, tetracycline, ciprofloxacin, roxithromycin, and amoxicillin are showed in Figure 3" Can we use Perovskite catalysts with AOPs for removing of antibiotics in wastewater? If yes, could you explain the mechanism?

Comment 18: "Through the catalytic activity of microorganisms, MESs obtain electrons from available organic matter or inorganic substances. Depending on the partial potential in the MESs, microorganisms can act as electron donors or electron acceptors" Can humic substances and Fe(III) affect the action of these microorganisms as electron donors or electron acceptors? Could you explain the possible use of extremophilic microorganisms in MESs in the bioremediation of antibiotics?

Note 9: "Xue et al. reported that more than 85% of sulfamethoxazole was degraded within 60 hours, and sulfamethoxazole could be completely degraded into less harmful substances, like alcohols and methane, and the ARGs produced in MESs were much lower than that in the traditional sewage treatment plants" Consider that with the presence of microorganisms (especially through electron syntrophy) and electron acceptors in the presence of organoelemental peroxides, the methane oxidation and antibiotic degradation ration will be affected.

Comment 19: "As an ecological sewage treatment technology, the constructed wetland has the characteristics of low operation cost, good quality of effluent, and easy maintenance. It can effectively remove antibiotics in wastewater based on the matrix adsorption and interception, the plant absorption and degradation, and the microbial decomposition" Elucidate the changes in physicochemical characteristics of the sewage effluent which involves antibiotics under constructed wetland technology treatment.

Comment 20: References:

Note 10: "Antibiotics are widely used in the treatment areas of diseases, such as bacterial infections in humans and animals, greatly reducing the mortality and morbidity of infectious diseases caused by bacteria and pathogens" Add the following reference:

Ventola, C Lee, 2015. The antibiotic resistance crisis: part 1: causes and threats. Pharmacy and Therapeutics 40(4): 277–283.

Note 11: "More importantly, antibiotics in the environment may induce the generation and spread of antibiotic resistance bacteria (ARB) and antibiotic resistance genes (ARGs), which can spread to the organisms and the humans through horizontal gene transfer routes in the direct contact or the food chain, leading to the resistance to antibiotics in ecosystem" Add the following reference:

2018.Research progress on distribution, migration, transformation of antibiotics and antibiotic resistance genes (ARGs) in aquatic environment. Critical Reviews in Biotechnology 38(8), 1195-1208.

Note 12: "The antibiotics are even found in groundwater. Draining from soil, infiltration from sediment, and introduction through the exchange between surface water and groundwater are three main sources of antibiotics in groundwater" Add the following reference:

2021.Inconsistent seasonal variation of antibiotics between surface water and groundwater in the Jianghan Plain: Risks and linkage to land uses. Journal of Environmental Sciences 109, 102-113.

Note 13: "Residual antibiotics in the environment pose a series of toxicological effects and ecological risks" Add the following reference:

2013. Toxicity of five antibiotics and their mixtures towards photosynthetic aquatic organisms: Implications for environmental risk assessment. Water Research 47(6), 2050-2064.

Note 14: "Due to high investment and operating cost, membrane filtrations are mostly used in drinking water treatment facilities, and rarely used in the antibiotics wastewater treatment" Add the following reference:

2018. Advanced highly polluted rainwater treatment process. Journal of Urban and Environmental Engineering 12(1), 50-58.

Note 15: "The non-biological degradation includes photolysis, hydrolysis, oxidative degradation, ionizing radiation degradation, and so on, and the biodegradation includes the microbial degradation and the plant degradation" Add the following reference:

2019.Degradation mechanisms of oxytetracycline in the environment. Journal of Integrative Agriculture 18(9), 1953-1960.

Note 16: "The direct photolysis refers to the process in which antibiotics directly absorb photons, which causes the molecules to the excited state by the light energy transformation, and subsequently form products by the bond breaking or the structural rearrangement" Add the following book chapter:

2007. Transformation of pharmaceuticals in the environment: Photolysis and other abiotic processes. Analysis, Fate and Removal of Pharmaceuticals in the Water Cycle, Chapter 3.2, Vol.50, pp.361-385.

Note 17: "The effect of membrane filtration depends on the physicochemical properties of solution and compounds, as well as the material properties of membrane" Add the following reference:

2007. Effects of membrane fouling on the nanofiltration of pharmaceutically active compounds (PhACs): Mechanisms and role of membrane pore size. Separation and Purification Technology 57(1), 176-184.

Note 18: "Membrane filtration can effectively remove macromolecular compounds including antibiotics" Add the following reference:

2015. Coupling of membrane filtration and advanced oxidation processes for removal of pharmaceutical residues: A critical review. Separation and Purification Technology 156, 891-914.

Note 19: "At present, lots of research has focused on the composite nanofiltration membranes, such as silver-mixed nanomaterials, multi-walled carbon nanotubes, and titanium dioxide" Add the following references:

2019.Developed greener method based on MW implementation in manufacturing CNFs. International Journal of Nanomanufacturing 15(3), 269-289.

Note 20: "As the photoenergy carriers, the photosensitizer can alter the photostability of compounds to accelerate photolysis" Add the following reference:

2020.An overview of photocatalytic degradation: photocatalysts, mechanisms, and development of photocatalytic membrane. Environmental Science and Pollution Research 27(3), 2522-2565.

Note 21: "Photolysis is one of the important ways to degrade antibiotics in the environment" Add the following reference:

2020.UV photolysis as an efficient pretreatment method for antibiotics decomposition and their antibacterial activity elimination. Journal of Hazardous Materials 392, 122321.

Note 22: "Antibiotics that can be hydrolyzed are mainly these that are soluble in water, and the degree of hydrolysis is depended on the features of different antibiotics" Add the following reference:

2020.THE STUDY OF PHYSICAL AND CHEMICAL PROPERTIES OF WATER-SOLUBLE POLYMER REAGENTS AND THEIR COMPATIBILITY WITH ANTIBIOTICS. Rasayan Journal of Chemistry 13(03), 1417-1423.

Note 23: "The factors affecting the hydrolysis of antibiotics are mainly pH value and temperature" Add the following reference:

2008. Effects of Ionic Strength, Temperature, and pH on Degradation of Selected Antibiotics. Journal of Environmental Quality37(2), 378-386.

Note 24: "Biodegradation is an important pathway for antibiotic degradation in the environment, chiefly including microbial degradation and plant uptake degradation" Add the following reference:

2020.Microbial degradation of tetracycline in the aquatic environment: a review. Critical Reviews in Biotechnology 40(7), 1010-1018.

Note 25: "There are lots of strains selected from the environment that can degrade antibiotics, such as photosynthetic bacteria, lactic acid bacteria, actinomycetes, yeast, fermentation filamentous bacteria, bacillus, subtilis, nitrated bacteria, and so on" Add the following reference:

2019.Plant growth promoting microbes: Potential link to sustainable agriculture and environment. Biocatalysis and Agricultural Biotechnology 21, 101326.

Note 26: "The main influencing factors include pH, temperature, oxygen content, and environmental media in the process that microbes degrade antibiotics" Add the following reference:

1997. Postexposure factors influencing the duration of postantibiotic effect: significance of temperature, pH, cations, and oxygen tension. Antimicrobial Agents and Chemotherapy 41(8), 1693-1696.

Note 27: "Firstly, the plants directly absorb organic pollutants and convert them into nontoxic metabolites accumulated in the tissues" Add the following reference:

2021. Evaluation of physicochemical characteristics and health risk of polycyclic aromatic hydrocarbons in borehole waters around automobile workshops in Southeastern Nigeria. Groundwater for Sustainable Development 14, 100615.

Note 28: "At present, the designed process of urban wastewater treatment plant effectively remove the conventional organic pollutants and salts, and the antibiotics are not within the target list of removing pollutants, resulting in its low removal efficiency" Add the following reference:

2021.Technological advancement for eliminating antibiotic resistance genes from wastewater: A review of their mechanisms and progress. Journal of Environmental Chemical Engineering 9(5), 106183.

Note 29: "It can be seen that urban sewage treatment plants have become an important point source for antibiotics to various environmental media" Add the following reference:

2019.Removal efficiencies of top-used pharmaceuticals at sewage treatment plants with various technologies. Journal of Environmental Chemical Engineering 7(5), 103294.

Note 30: "Biotechnology treatment is the degradation of antibiotics by organisms (mainly bacteria and fungi), which may also be accompanied by adsorption, hydrolysis, and photolysis" Add the following reference:

2020.Use of microalgae based technology for the removal of antibiotics from wastewater: A review. Chemosphere 238, 124680.

Note 31: "Plenty of factors can influence the treatment efficiency of AOPs, including the capacity of oxidant oxidation, the dose of oxidant, the action of catalyst, pH, pollutant concentration, and so on" Add the following reference:

2013. Solar photocatalytic oxidation and disinfection of municipal wastewater using advanced oxidation processes based on pH, catalyst dose, and oxidant. Journal of Renewable and Sustainable Energy 5(2), 023124.

Note 32: "Finally, ROSs further react with these intermediates to produce end products, such as carbon dioxide, water, and inorganic ions" Add the following reference:

2021.Comparative study of the biochemical response behavior of some highly toxic minerals on selenosis in rats. Revista de Chimie 72(2), 9-18.

Note 33: "However, due to the disadvantage of high operating cost, AOPs cannot be applied to the treatment of antibiotic wastewater on a large scale" Add the following reference:

2021.Comparative overview of advanced oxidation processes and biological approaches for the removal pharmaceuticals. Journal of Environmental Management 288, 112404.

Note 34: "The microbial communities in the constructed wetlands are better resistant to antibiotic stress, and it has been shown that the microbial structures in the constructed wetlands were more diverse compared with those of the conventional activated sludge" Add the following reference:

2017. Microbial community response during the treatment of pharmaceutically active compounds (PhACs) in constructed wetland mesocosms. Chemosphere 186, 823-831.

Note 35: "The physicochemical properties of antibiotics are one of the main reasons affecting the properties of the matrix adsorption, especially the water solubility and charge property" Add the following reference:

2015. Adsorption behavior of antibiotic in soil environment: a critical review. Frontiers of Environmental Science & Engineering 9(4), 565-574.

Note 36: "At the same time, it has been found that not only microalgae can remove antibiotics through biodegradation, but also their excreted photosensitive metabolites (such as extracellular polymers substances) can remove antibiotics by indirect photolysis" Add the following reference:

2022.Enhanced production of microalgae-originated photosensitizer by integrating photosynthetic electrons extraction and antibiotic induction towards photocatalytic degradation of antibiotic: A novel complementary treatment process for antibiotic removal from effluent of conventional biological wastewater treatment. Journal of Environmental Management 308, 114527.

Note 37: "Fourth, algae and their secretions may act as photosensitizers to induce indirect photolysis of organic pollutants" Add the following reference:

2021.Algae-induced photodegradation of antibiotics: A review. Environmental Pollution 272, 115589.

Note 38: "The removal mainly depends on the degradation or transformation of antibiotics by anaerobic or aerobic microorganisms" Add the following reference:

2013. Removal and degradation characteristics of quinolone antibiotics in laboratory-scale activated sludge reactors under aerobic, nitrifying and anoxic conditions. Journal of Environmental Management 120, 75-83.

Note 39: "In general, the removal rates of antibiotics are improved through the domestication and optimization of microbial populations" Add the following reference:

2020.Effect of carbon source on pollutant removal and microbial community dynamics in treatment of swine wastewater containing antibiotics by aerobic granular sludge. Chemosphere 260, 127544.

Note 40: "However, the biological treatment does not completely remove the antibiotics from the wastewater" Add the following reference:

2018.Antibiotic resistance and wastewater: Correlation, impact and critical human health challenges. Journal of Environmental Chemical Engineering 6(1), 52-58.

Reviewer 3 Report

The authors intended to review the processes and technologies for the degradation of antibiotics. The manuscript is properly structured, but a significant improvement is still required for further consideration.

# The review paper lacks literature from recent years (2020-2022). It should be at least over 20% of the total references, or the novelty of a review paper would be doubted.

# Please check the citation form in the entire manuscript, e.g., in lines 52 and 77.

# “Antibiotics that stay in the human can interact with microbes in the human”. Check the language use of the sentence. What is the meaning of “in the human”, the human body or the human environment?

# Use one paragraph to specify the purpose and highlight the review works contained in the manuscript.

# Pay attention to the paragraph structure of the text. E.g., in Lines 91-92 and 166-167, how could a sentence be a single paragraph in an academic paper?

# It would be much better to provide equations or figures to visualize the mechanism of different reaction processes, e.g., photolysis, hydrolysis…

# Table 2, how about the antibiotics in the wastewater of mainland China? It should be included in the table.

In each subsection of section 3, the authors should include a table to summarize recent research using the technology for antibiotics degradation. Moreover, the main advantages and drawbacks of each technology should be highlighted.

# The authors mentioned that it would be better to combine different processes for antibiotic removal. This reviewer believes that many existing studies have investigated combined processes for antibiotic removal. The paper should include a single section to review the combined process, and the authors should compare and analyze the results of existing combined technologies.

Reviewer 4 Report

This paper proposes a review of the treatments aimed at removing antibiotics from water.

The discussion is interesting and well-constructed. English needs to be revised because there are sparse typos and mistakes (i.e. emergent pollutant; eliminate rate).

Before publication the following comments should be considered:

11)      The authors have not considered including the electrochemical advanced oxidation treatments which represent a wide and effective class of processes based on the synergistic action of oxidants both externally added or electrogenerated with electrolysis (for example those ozone-based or those combining anodic oxidation with UV light or sonication). This choice is unclear since both electrochemical treatments and advanced chemical oxidation treatments are instead mentioned.

2)      the number of citations is not balanced between the various paragraphs and a greater number of articles from the last two years should be included.

32)      to understand the viability of a process and the possibility of scale-up, it is also necessary to mention the costs associated with the treatments

43)      the abstract and the conclusions report too similar sentences. Please, avoid repetitions.